# Influence of Tibetan Plateau snow cover on East Asian atmospheric circulation at medium-range time scales

Wenkai Li[1,2,3], Weidong Guo[2,3], Bo Qiu[2,3], Yongkang Xue[4], Pang-Chi Hsu[1] & Jiangfeng Wei[1]

The responses of atmospheric variability to Tibetan Plateau (TP) snow cover (TPSC) at seasonal, interannual and decadal time scales have been extensively investigated. However, the atmospheric response to faster subseasonal variability of TPSC has been largely ignored. Here, we show that the subseasonal variability of TPSC, as revealed by daily data, is closely related to the subsequent East Asian atmospheric circulation at medium-range time scales (approximately 3–8 days later) during wintertime. TPSC acts as an elevated cooling source in the middle troposphere during wintertime and rapidly modulates the land surface thermal conditions over the TP. When TPSC is high, the upper-level geopotential height is lower, and the East Asia upper-level westerly jet stream is stronger. This finding improves our understanding of the influence of TPSC at multiple time scales. Furthermore, our work highlights the need to understand how atmospheric variability is rapidly modulated by fast snow cover changes.

[1] Key Laboratory of Meteorological Disaster, Ministry of Education (KLME)/Joint International Research Laboratory of Climate and Environment Change (ILCEC)/Collaborative Innovation Center on Forecast and Evaluation of Meteorological Disasters (CIC-FEMD), Nanjing University of Information Science & Technology, 210044 Nanjing, China. [2] Institute for Climate and Global Change Research, School of Atmospheric Sciences, Nanjing University, 210023 Nanjing, China. [3] Joint International Research Laboratory of Atmospheric and Earth System Sciences, Nanjing University, 210023 Nanjing, China. [4] Department of Geography and Department of Atmospheric and Oceanic Sciences, University of California, 90095 Los Angeles, CA, USA. Correspondence and requests for materials should be addressed to W.G. (email: guowd@nju.edu.cn)

The lower boundary conditions of the atmosphere, including land surface and sea surface conditions, force the atmosphere and can serve as indicators of atmospheric circulation. One of the most important lower boundary conditions is snow cover. Snow cover, a crucial component of the cryosphere, affects the global climate system via the sensitivity of the radiation balance to the high-albedo characteristics of snow and the energy allocation involved in the melting of snowpack[1–16]. The Tibetan Plateau (TP) is the highest plateau in the world and is known as the third pole[17]. Thermal forcing by the TP influences weather and climate[18–23]. Due to its high elevation, the TP is colder than other surface regions at the same latitude and has much more snow cover. The variability of TP snow cover (TPSC) influences the land surface thermal conditions and thus influences the weather and climate in and around the TP[1,7,9,11–13,15,24–29].

The response of atmospheric variability to TPSC at seasonal, interannual and decadal time scales has been extensively investigated[7,9,11–13,15,24–29]. TPSC is regarded as an important indicator for climate prediction. Generally, climate prediction methods use the seasonal average TPSC conditions. For example, the winter or spring mean state of TPSC has been used as an indicator of summer monsoon precipitation anomalies over China[7,13,15,26,27]. However, the faster subseasonal variability of TPSC revealed by daily snow cover data and its effects on the atmosphere have largely been ignored. Recent studies have shown that accurate snow cover initialization can improve subseasonal and seasonal forecasts[30–33], implying that snow cover is a potential indicator for forecasts at shorter time scales. The internal atmospheric variabilities with high-frequency are significant over the TP[34]; and TPSC may serve as part of the source of these variabilities. A better understanding of the atmospheric effects of TPSC at multiple time scales, including shorter time scales, allows us to understand all aspects of atmospheric variability.

This study demonstrates that the subseasonal variability of TPSC, which has previously received little attention, is a dominant variability of TPSC and is nonnegligible. We further investigate the atmospheric effects caused by such fast subseasonal variability of TPSC. The anomalous TPSC rapidly influences land surface sensible heat flux over the TP, which directly affects subsequent local and downstream upper-level atmospheric circulation. This process of response of downstream atmospheric circulation are at an approximate 3–8 days lag, which falls in the medium range. Hopefully, this work will bring attention to the fast subseasonal variability of TPSC and its effects on the atmosphere.

## Results

### The subseasonal variability of Tibetan Plateau snow cover. We first investigated whether the subseasonal component of TPSC is significant. Here, the subseasonal variability means the variability with a period shorter than a season (e.g., 120 days). We analyzed the subseasonal variation of TPSC during wintertime by using daily snow cover data (Fig. 1a). Most areas have standard deviation greater than 20%. Meanwhile, some areas have values greater than 28% over the central and eastern TP. We also note that some parts of the western TP (marked by black oval in Fig. 1a) and the Himalayas have smaller standard deviations than areas over the central and eastern TP. The western TP and Himalayas are covered by snow during more than 90% of wintertime (Fig. 1b). The high climatological mean of the wintertime snow cover in these areas leads to smaller standard deviations because they are covered by snow almost all winter; however, these areas only represent a small percentage of the whole TP. For regions across almost the entire TP, especially the central and eastern TP, strong subseasonal variability generally occurs.

To measure the regional variability of TPSC, a TPSC index (TPSCI) was defined (see Methods). The TPSCI is directly proportionate to the snow-covered area over the TP. We decomposed the TPSCI (with the annual cycle removed; Supplementary Fig. 1a) into a subseasonal component (Supplementary Fig. 1b) and an interannual-to-decadal component (Supplementary Fig. 1c). The subseasonal component has a greater variability than the interannual-to-decadal component, especially during wintertime. The subseasonal component explains 65.8% of the total non-seasonal TPSCI variability, while the interannual-to-decadal component explains the remaining part. We also selected a representative case to illustrate rapid snow cover changes (Supplementary Figs. 2, 3). Supplementary Fig. 2 shows the TPSCI (red line) and its subseasonal anomalies (blue line) during the cold season of 2012/2013, while Supplementary Fig. 3 shows the spatial snow cover on selected dates. Although the climatology of the TPSCI (i.e., annual cycle) increases during this period (gray line), the TPSCI shows both increasing and decreasing changes in this season. For example, the TPSCI is 53.1% on 2012-12-29 (Supplementary Fig. 3a), which is higher than its climatology. Then, the TPSCI decreases over the following 7 days. The TPSCI decreases to 24.2% on 2013-01-06 (Supplementary Fig. 3b). Most of the TP area is without snow cover, except for some parts of the western TP and the Himalayas. Then, the TPSCI increases over the month and reaches 63.9% on 2013-01-20 (Supplementary Fig. 3c). Subsequently, the TPSCI decreases again and is close to its climatology on 2013-01-27 (Supplementary Fig. 3d). During the period from 2013-01-06 to 2013-01-27 (about half a month), the change in the TPSCI was up to 39.7%. The obvious subseasonal variation, such as the 39.7% change in the snow-covered area ($\sim 1.1 \times 10^6$ km$^2$) encompasses a broad area. The fast variability of snow cover over such a broad area may induce the atmospheric variability at a rapid time scale.

The above results demonstrate that the subseasonal variability of TPSC, which has previously received little attention, is dominant and nonnegligible, while seasonal averages may neglect this significant variability. It is thus valuable to further investigate the atmospheric effects caused by such fast variability.

### Influence of Tibetan Plateau snow cover on atmosphere. We found that the subseasonal variability of TPSC influences subsequent upper-level regional atmospheric circulation, especially the East Asia upper-level westerly jet stream (EAJS), during wintertime. The lead-lag relationship between TPSC and EAJS falls into medium-range time scales (~3–8 days). Both a statistical analysis based on ERA-interim reanalysis data and numerical experiments were performed to reveal this relationship.

The EAJS is one of the most distinct sources of atmospheric variability over East Asia[35]. The wintertime EAJS over subtropical regions varies across a wide range of time scales and is closely associated with many features of East Asia weather and climate[36–39]. Therefore, understanding variations in the of EAJS is important for accurate forecasting. The climatological mean of the wintertime upper-level (300-hPa) zonal wind reflects strong westerly winds over East Asia (Contours in Fig. 2). There is a strong westerly wind belt over the northwestern Pacific and Japan, with a zonal wind greater than 50 m s$^{-1}$. This upper-level westerly wind has the highest of global upper-level zonal wind values and is known as the jet core[35].

Composites were performed with respect to the TPSCI to assess the difference between positive and negative extreme cases of snow cover (see Methods). Figure 2 shows the composite evolution of the difference in 300-hPa zonal wind (U300). The composites show a lead–lag relationship between TPSC and

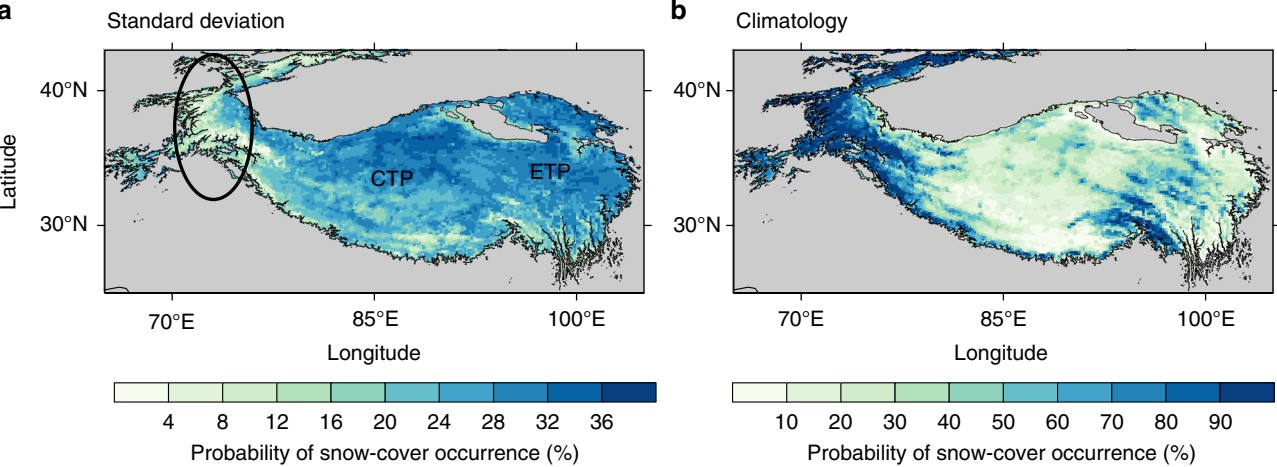

**Fig. 1** Standard deviation and climatology of wintertime Tibetan Plateau snow cover based on daily data. **a** Standard deviation of the 120-day high-pass filtered anomalous probability of snow-cover occurrence over the Tibetan Plateau (TP) during wintertime. **b** Climatological probability of snow-cover occurrence over the TP during wintertime. The unit is %. The black oval and text strings in **a** mark western TP, central TP and eastern TP from left to right. Gray areas indicate areas with altitudes less than 3000 m. The map of TP was created by using topographic data from Global Relief Model data of ETOPO1 (doi:10.7289/V5C8276M)

U300. These composites represent the responses of the atmosphere to increased TPSC. First, there is a significantly anomalous U300 with a north–south dipole structure around the TP at lag 1–2 days (Fig. 2a). Then, the persisting anomalous dipole structure of the U300 moves eastward (Fig. 2b). The positive anomalous westerlies reach the EAJS core region at lag 5–6 days (Fig. 2c). The composite of the anomalous U300 near the south of the Japanese archipelago (near 20–30°N, 120–150°E) is significantly positive, with values of ~5 m s$^{-1}$ (anomalous westerlies). Meanwhile, the anomalies still show a north-south dipole structure. There are negative anomalies near 40–50°N and 120–150°E (anomalous easterlies). Then, the U300 anomalies keep propagating eastward and weaken (Fig. 2d).

The above lead–lag composites indicate that the U300 varies with TPSC. Note that the composites contain signals not only from the lower boundary conditions (i.e., snow cover) but also from the internal atmospheric variability. In other words, both the TPSC-forced atmospheric response and the initial internal atmospheric variability may contribute to the TPSC variability and the downstream jet variability. To isolate the TPSC-forced atmospheric response (feedbacks exerted by snow) and to demonstrate that the composites contain causality of relationship and to further reveal the mechanism of interest, we performed a series of numerical experiments (see Methods). We carried out positive anomalous TPSC experiments (ExpPOS) and negative anomalous TPSC experiments (ExpNEG). The difference between the ExpPOS and ExpNEG represents the atmospheric response to the increased TPSC, while the response to the decreased TPSC has the opposite sign of this difference.

The response of the U300 in the numerical experiments is similar to that in the composite. This result clearly shows that a dipole structure of the U300 response originates from the TP (Fig. 3a). Then, the anomalous U300 moves eastward (Fig. 3b). The most evident response of the U300 over the EAJS core occurs after a 5–6 days lag (i.e., 5–6 days after the model initial time; Fig. 3c), which is consistent with the observed response (Fig. 2c). The U300 response is significantly positive near the EAJS core region. Then, the U300 anomalies continue to propagate eastward and weaken (Fig. 3d). The numerical experiments reproduce the dipole structure of the U300 anomalies in response to TPSC variability found in the reanalysis. However, there is still a slight difference between the reanalysis composites and the numerical

experiments response signal over the TP and the Indian subcontinent. The intensity of the zonal wind response in the numerical experiments is not as strong as that in the reanalysis composites. The slight difference between the reanalysis and model experiments could be caused by many factors, including model biases, model lateral boundary forcing constraints, which limit the model response to TPSC changes, and the limited effect of the initial change in snow cover in the models (only over TP) compared to the real cases (different everywhere). Despite these biases, the numerical experiments could reproduce the major patterns and evolution of the U300 anomalies.

The response of upper-level atmospheric circulation extends over East Asia, especially over the EAJS region. An EAJS index (EAJSI) was defined to measure the variability of the EAJS. The EAJSI was calculated by averaging the U300 speed over the EAJS core region (magenta box in Fig. 2a). This index is directly proportional to the intensity of the EAJS. The composite of the EAJSI with respect to TPSC is shown in Fig. 4 (black line in Fig. 4a). The EAJSI shows a delayed variation with respect to TPSC (EAJSI lags TPSC). The EAJSI increases after a lag of 0–5 days and peaks after a 6-day lag. Then, the EAJSI anomaly decreases. The EAJSI from the ExpPOS numerical experiments relative to that from the ExpNEG experiments isolates the response of the EAJSI to TPSC (blue line and light blue shading in Fig. 4a). The EAJSI in the numerical experiments also increases in the first 5 days after the initial time of the model. Specifically, the largest EAJSI response also occurs after a 6-day lag. Then, the response weakens. Thus, the EAJSI has a delayed response to TPSC, which makes TPSC can acts as an indicator of EAJS at medium-range time scales.

**Land surface energy budget and its atmospheric effects.** The numerical experiments suggest that the observed relationships at medium-range time scales can be reproduced in the model. Snow cover influences the land surface albedo[40]. By checking the land surface energy fluxes over the TP between ExpPOS and ExpNEG, we found that TPSC influences both the albedo and surface energy (Table 1). The snow cover condition over the TP in ExpPOS and ExpNEG is the only difference at the initial time step. The difference in albedo between ExpPOS and ExpNEG is 0.37 on the first day in the numerical experiments. The increased

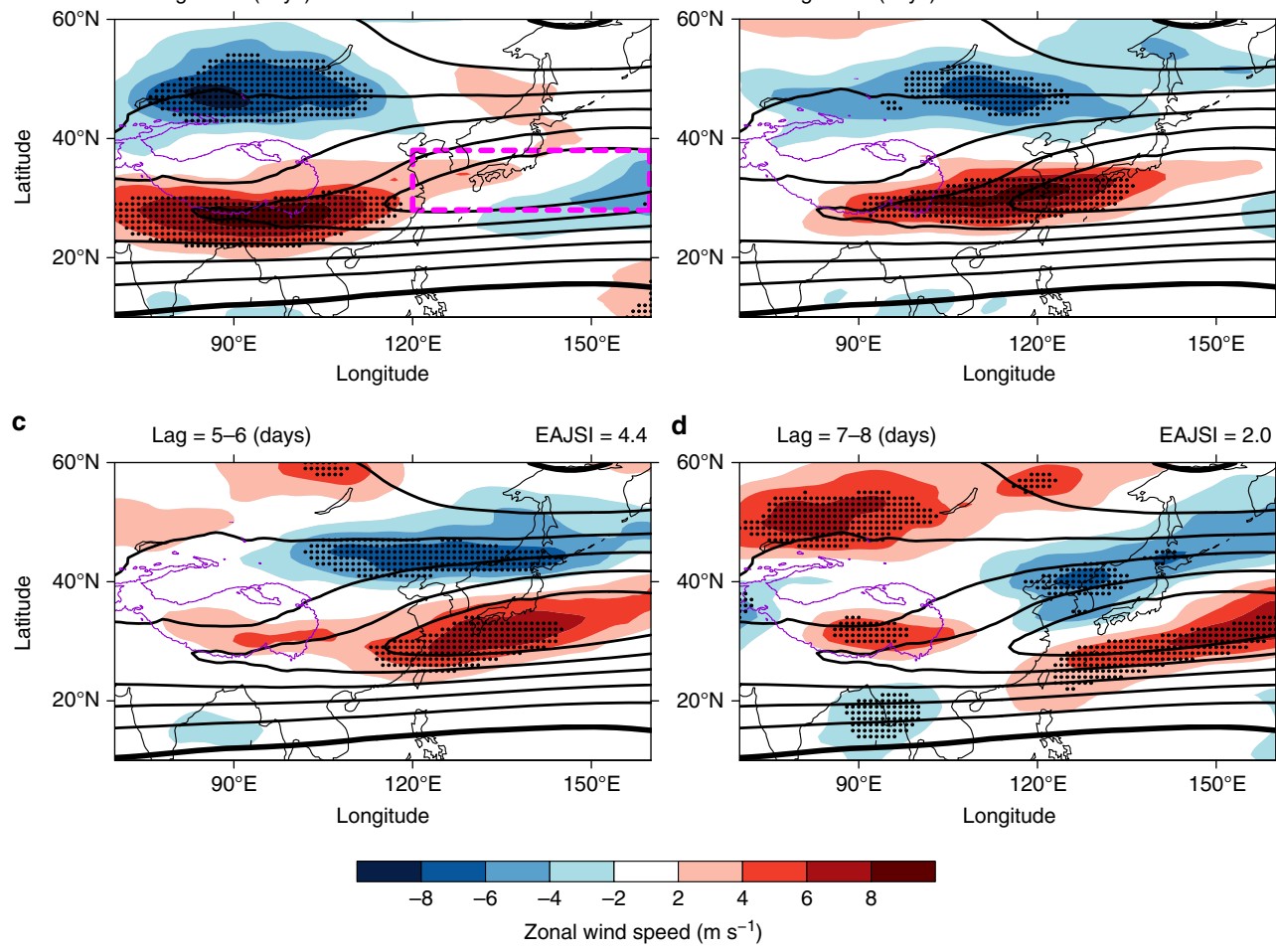

**Fig. 2** Response of the regional upper-level zonal wind to the subseasonal variability of Tibetan Plateau snow cover in observations. Shadings show the composite of the 300-hPa zonal wind with respect to the Tibetan Plateau snow cover index at lag **a** 1−2 days, **b** 3−4 days, **c** 5−6 days, and **d** 7−8 days. The unit is m s$^{-1}$. The purple contour marks the regions of the Tibetan Plateau with altitudes higher than 3000 m. Stippled regions mark composites with significance at the 99% level (two-side Student's $t$ test; see Methods). A lag of $n$ in the left title of each plot indicates that the 300-hPa zonal wind lags the Tibetan Plateau snow cover index by $n$ days. The right title of each plot indicates the East Asia upper-level westerly jet stream index for each plot. The black contours show the climatological mean of the wintertime 300-hPa zonal wind. The contour interval is 10 m s$^{-1}$. Contours with negative values are not shown. The zero contour is plotted with a bold black line. The magenta dashed rectangle in **a** shows the subdomain of the East Asia upper-level westerly jet stream core region (28–38°N, 120–160°E). The map of Tibetan Plateau was created by using topographic data from Global Relief Model data of ETOPO1 (doi:10.7289/V5C8276M)

albedo strongly increases the upward-reflected shortwave radiation. This difference in solar surface energy leads to a decrease in the absorbed solar radiation. Thus, the net shortwave radiation is decreased, while the response of the net longwave radiation is much smaller than that of the net shortwave radiation. The decreased absorbed solar radiation is mainly emitted by the land surface as sensible heat flux. Apart from the snow-albedo effect, snow cover insulates the overlying air from the soil layer below (snow-thermodynamic effect). The snow-thermodynamic also reduces the sensible heat flux at land surface. In contrast, the latent heat flux response is low. The overall responses of surface energy to the increased/decreased TPSC lead to anomalous cooling/heating effects, due to the snow-albedo effect and snow-thermodynamic effect.

The persistence of the snow anomaly signals in the observational and numerical experiments are shown in Fig. 4b. The composite at a 0-day lag of the TPSCI in the Interactive Multi-Sensor Snow and Ice Mapping System (IMS) analysis is 24.0%. In addition, the difference between the TPSCI of the ExpPOS and

ExpNEG at the first day of simulation is 24.5%, which is almost equal to that of the composite because the experimental design is based on the composite of TPSC. Then, the TPSCI anomaly in both the observational and numerical experiment composites decreases with lag days (Fig. 4b). The results show that the observed snow anomaly signal persists up to approximately 1 week and shows a decreasing tendency. The numerical experiments can reproduce both the persistence of the snow anomaly signal and the decreasing tendency within 6 days. The observational TPSCI anomalies fall sharply after 7 days, while the model that simulated TPSCI anomalies still shows a relatively slow decreasing tendency. The persistences of the TPSCI in ExpPOS and ExpNEG show similar results (Supplementary Note 1 and Supplementary Fig. 4). Although there is some bias 7 days after the initial date, the numerical experiments can generally reproduce both the persistence of the snow anomaly signal and the decreasing tendency at a medium range. A persistent surface energy response with a decreasing tendency is expected. Here, we take the sensible heat flux as a representative

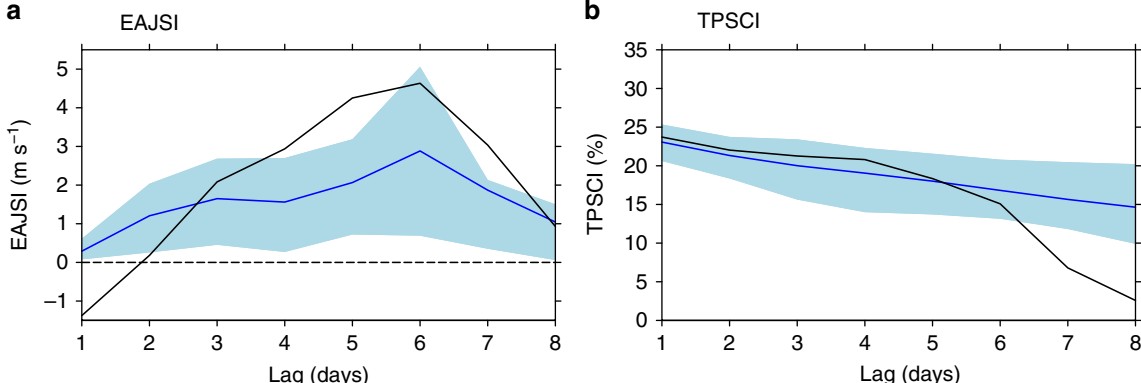

**Fig. 3** Response of the regional upper-level zonal wind to the subseasonal variability of Tibetan Plateau snow cover in numerical experiments. Responses of the 300-hPa zonal wind to Tibetan Plateau snow cover at lag **a** 1−2 days, **b** 3−4 days, **c** 5−6 days, and **d** 7−8 days. The unit is m s$^{-1}$. The purple contour marks the regions of the Tibetan Plateau with altitudes higher than 3000 m. A lag of $n$ in the title of each plot indicates the days by which the 300-hPa zonal wind lags the initial date of the model. The right title of each plot indicates the East Asia upper-level westerly jet stream index for each plot. The map of Tibetan Plateau was created by using topographic data from Global Relief Model data of ETOPO1 (doi:10.7289/V5C8276M)

**Fig. 4** The East Asia upper-level westerly jet stream index and Tibetan Plateau snow cover index from observations and numerical experiments. **a** The response of the East Asia upper-level westerly jet stream index to the subseasonal variability of Tibetan Plateau snow cover. **b** The persistence of the Tibetan Plateau snow cover index. The *x*-axis represents the number of days lagging the start of each event for the composites or the model initial date. The black line and blue line represent the reanalysis/analysis composites and numerical experiments, respectively. The light blue shading represents the range of the East Asia upper-level westerly jet stream index or Tibetan Plateau snow cover index between the 25th and 75th percentile of the numerical experiment ensembles. The unit is m s$^{-1}$ or %

**Table 1 The response of the Tibetan Plateau surface energy balance to the subseasonal variability of Tibetan Plateau snow cover in the numerical experiments[a]**

| Albedo | ↓SW | ↓LW | ↑SW | ↑LW | NetSW | NetLW | SH | LH |
|--------|-----|-----|-----|-----|-------|-------|-----|-----|
| 0.37 | 8.46 | −15.20 | 70.02 | −37.66 | −63.56 | 22.46 | −55.48 | 5.12 |

[a]The albedo and surface energy balance over the Tibetan Plateau (TP) on the first day (a lag of 1 day) in the numerical experiments. The unit of energy is W m$^{-2}$. ↓SW, ↓LW, ↑SW, ↑LW, NetSW, NetLW, SH and LH represent downward shortwave radiation, downward longwave radiation, upwards shortwave radiation, upward longwave radiation, net shortwave radiation, net longwave radiation, sensible heat flux and latent heat flux at the surface over the TP, respectively

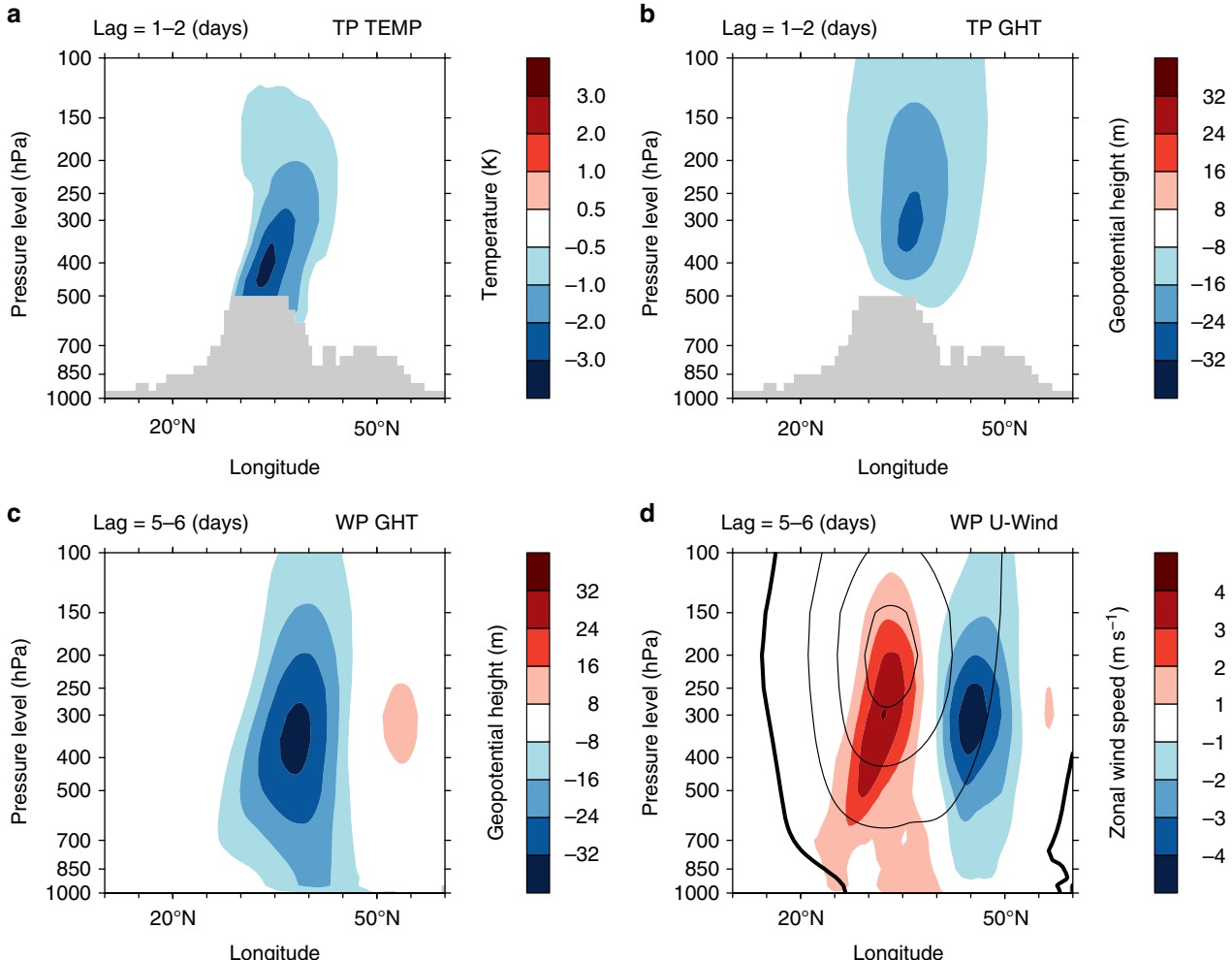

**Fig. 5** Response of the vertical profile of atmospheric temperature and circulation to the subseasonal variability of Tibetan Plateau snow cover in the numerical experiments. **a** Temperature responses over the Tibetan Plateau at lag = 1–2 day (averaged between 85 and 105°N; units: K). Gray areas indicate the mean terrain elevation. **b** As **a**, but for geopotential height (unit: m). **c** Geopotential height response over the western Pacific Ocean at lag = 5–6 days (averaged between 120 and 150°N; units: m). **d** As **c**, but for zonal wind (shading; unit: m s$^{-1}$). The black contours in **d** show the climatological mean of the wintertime zonal wind. Contour interval is 10 m s$^{-1}$. Contours with negative values are not shown. The zero contour is plotted with a bold black line

component of land surface energy to show the persistence of the land surface energy response (Supplementary Fig. 5). A strong negative response of sensible heat flux is caused by the snow-albedo effect for the first day, as discussed above. Then, the sensible heat flux increases (absolute value decreases) with lag days. At a 6-day lag, the sensible heat flux response is −22.8 W m$^{-2}$, implying that the surface energy can persist to a lag of approximately 1 week.

The TP acts as a cooling source during wintertime[19]. Decreases in the sensible heat flux strengthen the cooling effect. Due to the high elevation of the TP, the cooling effect directly affects the upper-level atmosphere (above a pressure level of 500 hPa). The

decrease in sensible heat flux for ExpPOS relative to ExpNEG is reflected by the upper-level atmospheric temperature. A substantial upper-level cooling response occurs over the TP. The temperature over the TP decreases rapidly after a 1–2 days lag (after the model initial time; Fig. 5a). The composite reanalysis of temperature also represents the upper-level cooling response (Supplementary Fig. 6). Such a cool response of TPSC is similar to that found in a previous study[31], but at seasonal time scales. The geopotential height changes accordingly (Fig. 5b). Geopotential height demonstrates a negative response over the TP. The maximum response of geopotential height occurs near the 500- to 200-hPa pressure level over the TP. The negative response of

the geopotential height over the TP moves eastwards with climatic westerly winds. The negative response of the geopotential height reaches the Japanese archipelago after an approximately 5–6 day lag (Fig. 5c). The zonal wind is in geostrophic balance with geopotential heights. Adaptive cyclone anomalies lead to the dipole structure of the zonal wind response, namely, the easterlies/westerlies to the north/south of the center of the geopotential height response (Fig. 5d).

Apart from the TP, snow cover is also widely distributed over Eurasia during boreal winter. Future studies on the possible rapid modulation by fast changes in snow cover over other parts of Eurasia are potentially valuable. This work also validates the sensitivity of the large-scale circulation simulated by numerical model to TPSC initialization. A more realistic TPSC initialization should help to decrease uncertainty in predictions or simulations. Although the model reproduces both the persistence of the snow anomaly signal and its tendency at the medium range, the model has some biases in simulating the sharp variations in TPSC after 7 days of observations. More works are necessary to further improve the snow and snow-atmosphere interaction processes in land surface models.

## Methods

**Data**. Two publicly available datasets are used in this study. (1) Daily snow cover data at a 24-km resolution are obtained from the Interactive Multi-Sensor Snow and Ice Mapping System (IMS) snow cover analysis[41]. (2) Daily averaged large-scale atmospheric circulation data are obtained from ERA-Interim[42]. The data range from November 1997 to February 2017. Each winter comprises 120 days from November 1 to February 28 of the following year. This study spans 20 extended winters (from 1 November 1997 to 28 February 2017).

**The definitions of snow cover and jet stream indices**. To measure the regional variability of TPSC, a TPSC index (TPSCI) is defined, which represents the percentage of the snow-covered area over the TP. The unit of the TPSCI is %. The TPSCI is calculated from the IMS snow cover analysis. The TPSCI is defined as $TPSCI = \frac{1}{n}\sum_{i=1}^{n} x_i \times 100\%$, where $x$ is the IMS snow cover analysis over the TP. If one grid point is covered by snow, $x = 1$; otherwise, $x = 0$. The grid points over the TP are defined as points at an altitude of >3000 m and within 25–43°N and 65–105°E. In total, according to these criteria, there are 7658 grid points over the TP. The TPSCI for the numerical experiments is also calculated. The TPSCI definition is the same as that used for the IMS analysis, but $x$ is the snow cover fraction in the model grid.

The EAJS index (EAJSI) is defined by averaging the 300-hPa zonal wind over the jet core region. The jet core region is the area in which the zonal wind is greater than 50 m s$^{-1}$ over the northwestern Pacific Ocean (magenta box in Fig. 2a).

**Validation of the snow cover analysis**. The IMS snow cover analysis, which is provided by the National Oceanic and Atmospheric Administration (NOAA), is an interactive system that is used to examine satellite images and other sources of data on snow cover and to generate maps of snow cover distribution[41,43,44]. The visible and infrared spectral data from the Polar Operational Environmental Satellites (POES) and Geostationary Orbiting Environmental Satellites were primarily used to generate snow cover data. Moderate Resolution Imaging Spectrometer (MODIS) imagery was used as well. In addition, ground weather observations from many countries were used. Since the visible and infrared data suffer from persistent cloud cover, which makes observations difficult, microwave products from SSM/I (Special Sensor Microwave Imager) and AMSR-E (Advanced Microwave Scanning Radiometer for EOS) are being used in the IMS product. The IMS system also includes the model output from Snow Data Assimilation System (SNODAS) and station-mapped products. The spatial resolutions of visible, infrared, microwave and SNODAS products used in the IMS System vary from 1 to 40 km.

The IMS product was manually created by NOAA NESDIS (The National Environmental Satellite, Data, and Information Service) satellite-product group analysts looking at all available satellite imagery, automated snow mapping algorithms, and other ancillary data. The IMS analysts use these multiple sources with different spatial resolutions within the interactive multisensor snow mapping system and re-gridded it to map snow at a 24-km spatial resolution. The analyst begins charting using the map from the previous day, then uses the satellite inputs accordingly. The IMS system allows for faster processing time to produce snow cover maps from satellite remote sensing data.

Reference[44] validated the IMS snow cover analysis by a comparison with ground-based measurements over the continental United States. They found that the IMS maps demonstrate a good correspondence with the ground-based measurements. The daily rate of agreement between the products mostly ranges between 80 and 90% in the Northern Hemisphere during the winter season,

when about a quarter to one-third of the continental US territory is covered with snow. Furthermore, they suggested that, when mapping snow cover, IMS analysts use the same technique and similar sources of data (e.g., satellite imagery, in situ data, and automated snow remote sensing products) over the whole Northern Hemisphere. Therefore, it is reasonable to assume that the accuracy of snow cover mapping over the mid-latitude region of Eurasia is similar to that over North America. Reference[45] found that the overall accuracy of IMS snow cover analysis is higher than 91% compared to station observations over the TP. Based on refs[44,45], the IMS snow cover analysis should be reliable to measure the overall variability of the TPSC. To test this hypothesis, we also evaluated the reliability of the IMS snow cover analysis over the TP. Here, we evaluated not only its overall accuracy but also the subseasonal variability of the TPSC in the IMS dataset.

To assess the quality of the IMS snow cover analysis over the TP, we used daily snow depth data collected from national meteorological stations of the China Meteorological Administration (CMA). The station data cover the period from 2000 to 2010, and we studied 55 stations over the TP region (Supplementary Fig. 7). The stations are mainly spread over the eastern TP. In the central and western TP, the weather observations are particularly lacking due to the bitter natural geographical environment and sparse population. To our knowledge, these station data are the only long-term operational observations including snow cover over the TP. It should be noted that CMA meteorological stations tend to be located in inhabited valleys. This may lead to differences between the station observations and satellite observations.

We evaluated the overall accuracy of the IMS snow cover analysis over the TP. We used an evaluation method similar that of refs[44,45]. The IMS pixel values (snow or no snow) were compared to the snow depth values of the station data. For a given station location that coincides with an IMS pixel, a "match" was tagged if the IMS pixel value was "snow" and the snow depth value from station data was "trace" or greater. Similarly, if the IMS pixel was classified as "land" and the station data recorded a snow depth of "0", a "match" occurred. A "mismatch" occurred if either the IMS pixel was "snow" and the station data showed a snow depth of '0' or the IMS pixel was classified as "land" and the station data showed a "trace" or greater snow depth. If the station value was missing for that day, it was tagged as "missing". Comparisons to the IMS data were not performed for those stations tagged as a "missing". The daily agreement between the IMS snow pixels and station data was calculated as the number of matching pixels divided by the sum of matching pixels ($N_{Matching}$) and mismatching pixels ($N_{Mismatching}$) multiplied by 100%:

$$OA = N_{Matching} / \left( N_{Matching} + N_{Mismatching} \right) \times 100\% \qquad (1)$$

where OA is the overall accuracy of IMS snow cover analysis. The overall accuracy was calculated for each day during the period of 2000–2010 (sample $N = 3650$). The overall accuracy over the TP based on our validation evaluation is 92%. Reference[45] found that the overall accuracy of IMS snow cover analysis is higher than 91% over the TP. Our evaluation is consistent with that of ref. [45]. Both our validation and ref. [45] suggest that the overall accuracy of IMS snow cover analysis over the TP demonstrate a good correspondence with ground-based measurements.

Our study focuses on the subseasonal variability of the TPSC. It is necessary to evaluate whether the IMS snow cover analysis can capture the subseasonal variability of the TPSC. Thus, the 'TPSCI' of both the station observations and IMS analysis for each winter during the period of 2000–2010 are compared. Note that here, the "TPSCI" is not the same as that defined previously. Here, the "TPSCI" only covers the 55 stations shown in Supplementary Fig. 7.

Also note that the snow condition observed at the stations is snow depth. Adopted from Biosphere-atmosphere Transfer Scheme (BATS)[46], the snow cover fraction ($f_{sno}$) is parameterized as a nonlinear function of snow depth ($h_{sno}$):

$$f_{sno} = h_{sno} / \left( 10 z_{g,0} + h_{sno} \right) \qquad (2)$$

where $z_{g,0}$ (= 0.01 m) is the ground roughness length. We estimate $f_{sno}$ for each station by using this parameterized function. The average of $f_{sno}$ for all 55 stations is the "TPSCI" of station observations.

The capture of the $f_{sno}$ for one station by IMS analysis is estimated using the following steps. First, the five IMS snow cover grids nearest to one station are selected. Then, the average of these five grids is regarded as the capture of one station by IMS analysis. Finally, the average of the $f_{sno}$ by IMS analysis for all 55 stations is regarded as the "TPSCI" for IMS analysis. Similar to the method in the main text, we apply a 120-day high-pass filter to remove the interannual and decadal components for these two "TPSCI" time series and focus on the subseasonal time scale.

The "TPSCI" of both stations and IMS for each winter during the period of 2000–2010 are compared (Supplementary Fig. 8). Overall, the IMS data capture the peaks of the "TPSCI" anomaly well for each winter during the period of 2000–2010. For example, the above-anomaly to below-anomaly transitions are captured correctly by IMS analysis (Supplementary Fig. 8i and Fig. 8j). The correlation coefficient (CC) between the two time series for each winter (the sample size $N = 120$) ranges from 0.39 to 0.79. The CC for the two time series for all 10 winters is 0.56 (the sample size $N = 1200$), indicating that the IMS snow cover analysis captures the subseasonal variability of the TPSC well.

The above evaluation shows that the IMS snow cover analysis can capture the general subseasonal variability of the TPSC. This dataset is valuable, especially for the TP, where observations are particularly lacking due to the bitter natural geographical environment and sparse population.

**Composite**. This study investigates the influence of TPSC on East Asia upper-level circulation using a composite analysis. Extreme anomalous TPSC events are selected according to the TPSCI from the IMS snow cover analysis. We first extract the subseasonal component of the TPSCI by using a 120-day high-pass filter to remove the interannual and decadal components. We then standardize the index. The extreme anomalous TPSC events are selected based on the following criteria. The TPSCI must be greater than 0.5 (less than −0.5) for 6 or more successive days. The largest (smallest) day among these 6 or more successive days is regarded as the "start day" of this one extreme positive (negative) anomalous TPSC event. The number of events for the composite are 45 and 64 for positive and negative events, respectively. We averaged the 45 (64) samples of positive (negative) anomalous event samples at the "start day". The difference between the average positive and negative anomalous events is the composite that lags the TPSCI by 0 days. The difference of the average atmospheric fields values for the following 1–8 days after the "start day" are the composites that lag TPSCI by 1 to 8 days. To focus on the subseasonal time scale, we applied a 120-day high-pass filter to remove the interannual and decadal components of the atmospheric fields.

**Significance tests**. The statistically significance test for the composites is based on the two-sided Student's $t$ test.

**Numerical model**. To confirm the conclusions from the composites and to further reveal the mechanism of interest, numerical experiments are performed. The numerical model used here is the Advanced Weather Research and Forecasting Model (WRF-ARW, version 3.9.1), which was developed by the National Center for Atmospheric Research (NCAR). The WRF-ARW has been applied to climate research, including studies of land–air interactions. The model domain is centered at 35°N and 115°E, covering an area of 9250 km (west–east) × 6350 km (south–north) with a horizontal resolution of 50 km. There are 28 levels in the vertical direction. The land surface parameterization scheme used in this study is the Noah-MP land surface model[47]. Noah-MP contains a multi-layer snow pack with liquid water storage and melt/refreeze capability and a snow-interception model describing loading/unloading, melt/refreeze, and the sublimation of the canopy-intercepted snow. Important physics options include the WRF single-moment 6-class microphysics scheme[48], the NCAR Community Atmosphere Model (CAM 3.0) spectral-band shortwave and longwave radiation schemes[49], the Yonsei University planetary boundary layer scheme[50], and the Kain–Fritsch convective parameterization scheme[51]. The WRF is driven by atmospheric and surface forcing data extracted from the National Centers for Environmental Prediction (NCEP) FNL Operational Model Global Tropospheric Analyses.

**Design of numerical experiments**. To test the response of atmospheric circulation to TPSC, two ensemble experiments are performed, namely, positive anomalous TPSC experiments (ExpPOS) and negative anomalous TPSC experiments (ExpNEG). The initial lower boundary condition TPSC is modified based on the IMS snow cover analysis. The target of the experimental design is to generate increased/decreased TPSC in ExpPOS/ExpNEG that are consistent with the IMS composite. The simultaneous composites of the daily TPSC from the IMS snow cover analysis were treated as "anomalies" (Supplementary Fig. 9a). The climatological TPSC plus half of the "anomalies" obtain the initial forcing for ExpPOS, whereas the initial forcing for ExpNEG is obtained by the climatological TPSC minus half of the "anomalies". Hence, the difference in TPSC between ExpPOS and ExpNEG are the 'anomalies' shown in Supplementary Fig. 9b. In the land surface scheme, the physical snow depth ($h_{sno}$) and snow water equivalent ($m_{sno}$) are the direct variables related to the snow cover fraction ($f_{sno}$). We still transformed the $f_{sno}$ to the $h_{sno}$ by using Eq. (2). The $h_{sno}$ in grids of IMS analysis was interpolated into model grid. The snow density ($\rho_{sno}$) is estimated as 100 kg m$^{-3}$ for ExpPOS to emulate the fresh snow increase in positive anomalous TPSC events, and the $\rho_{sno}$ is estimated as 350 kg m$^{-3}$ for ExpNEG to emulate the aged snow remaining in negative anomalous TPSC events. The $m_{sno}$ is then calculated from the modified $h_{sno}$ and the estimated $\rho_{sno}$. Except for $h_{sno}$ and $m_{sno}$, other variables at the initial time step remain unchanged. As a result, the initial $f_{sno}$ between ExpPOS and ExpNEG is the "anomalies" (Supplementary Fig. 9b).

The only difference at the initial time step of the model are the snow cover conditions over the TP for ExpPOS and ExpNEG. All other variables at initial time step remain unchanged. Then, the model is integrated freely. To eliminate the impacts of atmospheric variability embedded in the initial conditions, we carried out several runs for each experiment with different initial dates and conducted an ensemble mean. Both numerical experiments contain 12 members with different initial times. Here, we use the 1st, 11th, and 21st day in each month from November 2013 to February 2014 as the initial dates. Each member is run continuously for 15 days. The ensemble means of each member for different experiments are used for analysis. Since the initial date can be considered as random, the internal atmospheric variability in the initial time step could be largely smoothed out after the ensemble average is applied. Comparisons of ensemble mean results between ExpPOS and ExpNEG could efficiently reveal the TPSC-forced atmospheric responses. The numerical experiments can reproduce the persistence of the snow anomaly signal (Fig. 4b). Using the ensemble means of each member with different initiation times can remove the synoptic variability of the atmosphere. The difference between ExpPOS and ExpNEG is considered to represent the response or the sensitivity of the atmosphere to TPSC at medium-range time scales.

**Code availability**. All figures were produced using NCAR Command Language (NCL) version 6.4.0, an open source software free to the public, by UCAR/NCAR/CISL/TDD, https://doi.org/10.5065/d6wd3xh5. All the NCL scripts used in this study are available from the corresponding author upon request.

## Data availability

The IMS snow cover data are available at http://nsidc.org/data/docs/noaa/g02156_ims_snow_ice_analysis/. The ERA-interim data are available at http://apps.ecmwf.int/datasets/. The NCEP FNL data are available at https://rda.ucar.edu/datasets/ds083.2/. The WRF source codes can be obtained at http://www2.mmm.ucar.edu/wrf/users/download/get_sources.html. The topographic data from Global Relief Model data of ETOPO1 can be obtained at https://data.nodc.noaa.gov/cgi-bin/iso?id=gov.noaa.ngdc.mgg.dem:316. Other data that support the findings of this study are available from the corresponding author upon request.

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

## Acknowledgements

This research is supported by the Strategic Priority Research Program of Chinese Academy of Sciences, Grant No. XDA2006010104, Natural Science Foundation of China (41705056, 41475063), the Natural Science Foundation of Jiangsu Province (BK20170638), the Startup Foundation for Introducing Talent of NUIST, and the Priority Academic Program Development of Jiangsu Higher Education Institutions (PAPD). This work is also supported by Jiangsu Collaborative Innovation Center for Climate Change.

## Author contributions

W.G. led the overall scientific questions and designed the research. W.L. analyzed the data and drafted the manuscript. B.Q., Y.X., P.H., and J.W. gave conceptual advice. All authors contributed to the discussion of the results and to revising the manuscript.

## Additional information

**Competing interests:** The authors declare no competing interests.

