## [Peer Review File · Nature Communications]

Parts of this Peer Review File have been redacted as indicated to remove third-party material where no permission to publish could be obtained.

Reviewers' Comments:

Reviewer #1:

Remarks to the Author:

This paper examines the atmospheric response to short-term snow cover variability at the subseasonal time scale over the Tibetan Plateau during winter. This is achieved by compositing reanalyses upon a snow cover index derived from the largely satellite-based (IMS) snow cover data. The response to snowy events is also analysed in composite differences of 15-day ensemble experiments using a regional model (WRF), where the snow cover is added or removed. Both reanalyses and model tend to show a maximum response in the Asian jet core lagging the snow cover variations by 6 days.

Although I realise that there are a lot of supplementary figures already, the paper would gain by providing more evidence for the snow variations themselves in the IMS data, e.g. during a selected representative event. Meteorological conditions over the Tibetan Plateau tends to be dry and windy in winter. Are the snow cover variations located all over the Plateau or mostly in its western side and the western Himalayas (since the snow index is constructed using a broad longitudinal extent) ? Hence some evidence that the snow cover is high (and variable) over the entire TP should be provided. I suppose the authors refer to IMS data, because most snow analyses are unreliable over the TP due to large model biases. Some maps for a chosen representative event could be illustrative of the extent of rapid snow cover changes.

Are these synoptic events bringing rapid snow cover changes related to the well-known "Western Disturbances" (WD) that bring considerable snowfall over the Western Himalayas in winter, or not ? In that case, one might also question the causality of the relation between snow and jet stream variability. These travelling upper-tropospheric disturbances could be responsible for the snowfall, and sometime later, for some downstream jet variability. Some comments are warranted.

In the model experiments, these WD could be embedded in the lateral forcing of the WRF at the boundary of its domain. Presumably, taking composite difference of ensemble means should factor out this effect. Yet, some more data on the model simulations should also be incorporated to make the manuscript more convincing. The imposed snow cover in the model is quite thin (5 cm), and it would be interesting to show how it evolves and persists with time through the model simulations. For example, when the snow cover is artificially removed in experiment "ExpNoSnow", does it grow rapidly again due to the (inconsistent) atmospheric forcing?

The impact of snow initialisation on subseasonal forecasts (encompassing the medium-range time scale explored here with the WRF model) has been examined in a broader context in several recent papers with global models. I find the paper lacking those relevant references. In particular, a cold regional anomaly throughout the troposphere was also modelled in subseasonal forecasts during high snow depth conditions over Tibet by Senan et al. (2016), albeit in springtime.

Orsolini, Y.J., Senan, R., Balsamo, G., Doblas-Reyes, F., Vitart, D., Weisheimer, A., Carrasco, A., Benestad, R. (2013), Impact of snow initialization on sub-seasonal forecasts, *Clim. Dyn.*, 41:1969-1982.

Senan, R., Orsolini, Y.J., Weisheimer A., Vitart, F., Balsamo, G., Stockdale, T., Dutra, E., Doblas-Reyes, F., D. Basang, Impact of springtime Himalayan-Tibetan Plateau snowpack on the onset of the Indian summer monsoon in coupled seasonal forecasts, *Clim. Dyn.*, Vol. 47, Issue 9, pp 2709–2725, doi:10.1007/s00382-016-2993-y. (2016)

Jeong, J.H., H.W. Linderholm, S.-H. Woo, C. Folland, B.-M. Kim, S.-J. Kim and D. Chen (2013), Impact of snow initialization on subseasonal forecasts of surface air temperature for the cold season, *J. Clim.*, 26:1956-1972.

Few papers have examined this kind of response at the medium-range time scale, hence the results are quite novel. While the paper has merit, it should be more polished to be considered

for publication in Nature. Many of the statements could be made more precise. I recommend a major revision of the paper, before it is in an acceptable form for publication.

OTHER MAJOR COMMENTS

L123: the model composite and the reanalyses in Fig 1 are not completely similar. They differ over the TP itself, and over the Indian subcontinent. Some clarification is needed.

Figure 1 : More results should be integrated in the main body of the paper than the "snap-shot" 6-day lag shown in Figure 1. Some material from Figures S3 and S4, showing other lags, should be used to expand figure 1 in order to give the reader an overview of the subseasonal evolution.

L143 : the negative jet index response at short lags (lag 0 or 1) is not discussed. Nor is the strong zonal wind signal at lag 0 (seen in Figure S3), where the jet is enhanced on the southern side of the TP. Do the authors consider this as part of the near-instantaneous snow-induced response (as visible in temperature) ?

L164: I don't understand some points in the discussion of the surface energy balance. In presence of a thick snowpack, one expects also a cool surface anomaly (decreased sensible heat flux) due to the insulating property of snow (a thermodynamical and not radiative property). That should add to the same-sign temperature anomaly from the short-wave deficit. Some clarification is needed.

MINOR COMMENTS

L72-L74: much of that is a repeat of what was mentioned above.

L 210: why aren't the temperature composites from the reanalyses shown ?

Figure S9 : It would be useful to have the climatological winds in background as contours.

Reviewer #2:

Remarks to the Author:

Review of "Influence of the Tibetan Plateau snow cover on East Asian atmospheric circulation on medium-range time-scales" by Wenkai Li etc..

Recommendation: Major revision.

Overview:

This study investigated the response of the atmosphere to the subseasonal variability of the TPSC. The motivation of this study based on the extensively investigation and discussion of the response of atmospheric variability to the Tibetan Plateau (TP) snow cover (TPSC) at interannual to decadal time-scales, rather than at subseasonal time-scale. Results showed that subseasonal variability of the TPSC can serve as an indicator of East Asian atmospheric circulation on medium-range time-scales (approximately 3–7 days in advance). The motivation of this study is a good one and could be useful to the subseasonal prediction. I would recommend a major revision before acceptance.

Major comments:

1. The conclusion of this study is "The subseasonal variability of the TPSC influences the upper-level regional atmospheric circulations, especially the East Asia upper-level westerly jet stream (EAJS), during wintertime. The variability of the TPSC modulates the land surface thermal conditions over the TP, which acts as an elevated cooling source in the middle troposphere during wintertime. The upper-level geopotential height is affected by the TPSC and then the zonal wind. As a result, the EAJS is influenced". These mechanisms have long been well known to the community and can not be regarded as a new concept. Therefore, the discussion on the

mechanism has no new idea and lack of clearly illustration and elaboration in physical perspective.

2. The experiment design, mechanism illustration and conclusions in this study, actually emphasize the snow radiation cooling effect, these are known well by reader. An effective analysis of persistence of the snow cover cooling effects is needed. In fact, snow cover as a signal for predicts summer precipitation over Eastern China has been used for long time, but there is distinctly uncertainty.
3. Lack of rigorous verification with observations, increase/decrease albedo implies snow cover is uniform over TP, however, most of TP has no snow cover, even in wintertime. I am concerned about the experiment design and the possible mechanism that the authors did not fully addressed.
4. Similar to comment 3, the snow data used in this study comes from the Interactive Multi-Sensor Snow and Ice Mapping System (IMS). The reliability of this dataset is not clear, which could result in an unclear or even a wrong conclusion.
5. Uncertainty of the definition of TPSCI. TPSCI is defined with the IMS analysis data, As mentioned in comment 4, the datasets may have great uncertainties in representing the real snow cover distributions, thus, it is not clear whether the definition of TPSCI is reasonable or not, especially in the western TP.
6. Potential errors and misleadings of the model configuration and experiment design. First, WRF model is a collaborative achievement among NCAR, NCEP, ESRL, AFWA and NRL etc.; Second, the authors conducted the sensitivity experiments by roughly increasing or removing snow over the whole TP. Since the snow over the TP has great differences in spatial-temporal, previous studies have revealed and highlighted that snow cover in different region has much different impacts on the East Asian atmospheric circulation. Therefore, the experiment design is not reasonable and the corresponding results are not convincing to this reviewer.
7. The color bar in Fig. 1 is not consistent in each panel, which is misleading to this reviewer because the intensity and structure of regional upper-level zonal wind are not well reflected in the simulation.

Reviewer #3:

Remarks to the Author:

Influence of the Tibetan Plateau snow cover on East Asian atmospheric circulation on medium-range time-scales

By Li et al.

Recommendation: Major revisions

General Comments:

This study examines Influence of the Tibetan Plateau (TP) snow cover on East Asian atmospheric circulation on medium-range time-scale, which has been noticed by few previous works. They proposed the physical linkage is the initial surface sensible heating reduce, not surprise. The results are reasonable but need to clarify several issues. Therefore, I would recommend major revisions.

1. For the snow data used in this study. If this data is valid on daily time scale as well as over the TP? You need to show the evidence.
2. How the snow anomaly signal maintains in the observation and ExpSnow? How the surface energy change up to Lag=6?
3. For the physics of TP snow impact, what are the differences between medium-range and interannual time-scales? What's the new insight?
4. L30, Add 'during wintertime' after "in advance)". You didn't study other seasons.
5. L45, add 'at surface' before 'at the same lat' since it is not the case at pressure surface.
6. Fig. 1, the u300 anomaly is strong at lag=0, indicating the snow anomaly's contributor. Need

discussion here.

7. L169, Table 1, observation and numerical simulations. Which one? Only one number in Table 1 for one variable.

8. L249, great than 3000 m. How about 2000 m or 2500 m. If the results are sensitive to the altitude for snow index?

We appreciate the reviewer's valuable and thoughtful comments. We have incorporated these comments into the revised manuscript. The following is our point-by-point response. The blue italics are reviewer's comments. Our responses are provided in black.

Reviewer #1 (Remarks to the Author):

This paper examines the atmospheric response to short-term snow cover variability at the subseasonal time scale over the Tibetan Plateau during winter. This is achieved by compositing reanalyses upon a snow cover index derived from the largely satellite-based (IMS) snow cover data. The response to snowy events is also analysed in composite differences of 15-day ensemble experiments using a regional model (WRF), where the snow cover is added or removed. Both reanalyses and model tend to show a maximum response in the Asian jet core lagging the snow cover variations by 6 days.

Although I realise that there are a lot of supplementary figures already, the paper would gain by providing more evidence for the snow variations themselves in the IMS data, e.g. during a selected representative event. Meteorological conditions over the Tibetan Plateau tends to be dry and windy in winter. Are the snow cover variations located all over the Plateau or mostly in its western side and the western Himalayas (since the snow index is constructed using a broad longitudinal extent) ? Hence some evidence hat the snow cover is high (and variable) over the entire TP should be provided. I suppose the authors refer to IMS data, because most snow analyses are unreliable over the TP due to large model biases. Some maps for a chosen representative event could be illustrative of the extent of rapid snow cover changes.

Response:

Thanks for your comments. The reviewer concerns the snow cover variations over the whole Tibetan Plateau (TP). We agree with the reviewer that this is really an important issue. Here we represent the overall characteristic of the subseasonal variation of the TP snow cover (TPSC) during wintertime, and then we show some maps for a chosen representative event to illustrate the rapid snow cover changes.

1. Subseasonal variation of the TPSC

We analysed the subseasonal variation of TPSC during wintertime by using daily snow cover data (Fig. R1-1). Most areas have standard deviation greater than 20%. Meanwhile, some areas

have values greater than 28% over the central and eastern TP (CTP and ETP). We also note that some parts of the western TP (marked by black oval in Fig. R1-1a) and the Himalayas have smaller standard deviations than areas over the CTP and ETP. The western TP and Himalayas are covered by snow during more than 90% of wintertime (Fig. R1-1b). The high climatological mean of the wintertime snow cover in these areas leads to smaller standard deviations because they are covered by snow almost all winter; however, these areas only represent a small percentage of the whole TP. For regions across almost the entire TP, especially the CTP and ETP, strong subseasonal variability generally occurs.

Figure R1-1 | Standard deviation and climatology of wintertime TPSC based on daily data. (a) Standard deviation of the 120-day high-pass filtered anomalous probability of snow-cover occurrence over the TP during wintertime. (b) Climatological probability of snow-cover occurrence over the TP during wintertime. The unit is %. Grey areas indicate areas with altitudes less than 3,000 metres.

To confirm that the variability that is the focus of this study is over the entire TP, we further investigated the spatial TPSC variation related with the TPSCI by performing composite analysis (Fig. R1-2a). The composite analysis method is described in detail in the Methods section of this manuscript. Most areas over the TP have a composite value greater than 20%. High composite values greater than 30% and sometimes greater than 40% occur over the CTP and ETP. Generally, variations related with the TPSCI encompass almost the entire TP. This result is consistent with the total subseasonal variability.

We have incorporated the above discussion into the revised manuscript. Please see lines 57–66 in the revised manuscript. Fig. R1-1 and Fig. R1-2 are also Fig. 1 in the revised manuscript and S9 in the revised supplementary.

Figure R1-2 | The initial lower boundary condition TPSC based on the IMS snow cover analysis. (a) The simultaneous composites of the daily anomalous TP snow-cover probabilities for the IMS snow cover analysis. (b) The difference between the initial snow cover fraction of ExpPOS and ExpNEG. The unit is %. Grey areas indicate the areas with altitudes less than 3,000 metres.

2. A representative case

We selected a representative case to illustrate rapid snow cover changes (Fig. R1-3 and Fig. R1-4). Fig. R1-3 shows the TPSCI (*red line*) and its subseasonal anomalies (*blue line*) during the cold season of 2012/2013, while Fig. R1-4 shows the spatial snow cover on selected dates.

Figure R1-3 | The TPSCI and its subseasonal component in the cold season of 2012/2013. The *red line* shows the raw TPSCI (percentage of snow-covered area over the TP). The *blue line* shows the subseasonal component of the TPSCI. The *grey line* shows the annual cycle of the TPSCI. The *thin black line* shows the reference line of zero. The unit is %. (a)–(d) mark the selected dates on which the TPSC spatial distributions are shown in Fig. R1-4.

Although the climatology of the TPSCI (i.e., annual cycle) increases during this period (*grey line*), the TPSCI shows both increasing and decreasing changes in this season. For example, the TPSCI is 53.1% on 2012-12-29 (Fig. R1-4a), which is higher than its climatology.

Then, the TPSCI decreases over the following 7 days. The TPSCI decreases to 24.2% on 2013-01-06 (Fig. 1-4b). Most of the TP area is without snow cover, except for some parts of the western TP and the Himalayas. Then, the TPSCI increases over the month and reaches 63.9% on 2013-01-20 (Fig. 1-4c). Subsequently, the TPSCI decreases again and is close to its climatology on 2013-01-27 (Fig. 1-4d).

During the period from 2013-01-06 to 2013-01-27 (about half a month), the change in the TPSCI was up to 39.7%. The obvious subseasonal variation, such as the 39.7% change in the snow-covered area (approximately $1.1 \times 10^6 \text{ km}^2$) encompasses a broad area. The variability of snow cover over such a broad area may induce the atmospheric variability at a subseasonal time scale.

We have incorporated the above discussion into the revised manuscript. Please see Lines 28–54 in the revised supplementary. Fig. R1-3 and Fig. R1-4 are also Fig. S2 and Fig. S3 in the revised supplementary.

Figure R1-4 | The spatial distribution of TPSCI on selected dates. The white area shows the snow-covered grids from the IMS snow cover analysis, while the orange area shows the grids with no snow. The right title in each subfigure represents the percentage of snow-covered area over the TP.

Are these synoptic events bringing rapid snow cover changes related to the well-known “Western Disturbances” (WD) that bring considerable snowfall over the Western Himalayas in winter, or not? In that case, one might also question the causality of the relation between

snow and jet stream variability. These travelling upper-tropospheric disturbances could be responsible for the snowfall, and sometime later, for some downstream jet variability. Some comments are warranted.

Response:

[REDACTED]

The WDs embedded in the southward propagating midlatitude Subtropical Westerly Jet (SWJ) produce extreme precipitation over northern India and are further enhanced over the Himalayas due to orographic land-atmosphere interactions (Dimri et al. 2015; Madhura et al. 2015). Fig. R1-5 (Source: Fig. 6 in *Madhura et. al.* 2015) shows the pattern of precipitation related to WDs. The figure shows that precipitation anomalies gradually develop and propagate eastward along

with the WDs. The precipitation related to WDs occurs mainly over the western TP and Himalayas. As discussed in response to your comment #1, some parts of the western TP (marked by the black oval in Fig. R1-1a) and Himalayas have a smaller standard deviation than the CTP and ETP. The high climatological mean of wintertime snow cover in these areas leads to a smaller standard deviation because they are covered by snow almost all winter. The high climatological mean may be due to snowfall related to WDs. However, areas with large subseasonal variability mainly occur over the CTP and ETP. We think that the WDs may not be a dominant factor modulating the subseasonal variability of TPSC. Even so, additional work on the relation between the WDs and TPSC is expected in the future.

References

- Dimri, A. P. *et al.* Western Disturbances: A review. *Reviews of Geophysics* **53**, 225–246, doi:10.1002/2014RG000460 (2015).
- Madhura, R. K., Krishnan, R., Revadekar, J. V., Mujumdar, M. & Goswami, B. N. Changes in western disturbances over the Western Himalayas in a warming environment. *Clim. Dyn.* **44**, 1157–1168, doi:10.1007/s00382-014-2166-9 (2015).

In the model experiments, these WD could be embedded in the lateral forcing of the WRF at the boundary of its domain. Presumably, taking composite difference of ensemble means should factor out this effect. Yet, some more data on the model simulations should also be incorporated to make the manuscript more convincing. The imposed snow cover in the model is quite thin (5 cm), and it would be interesting to show how it evolves and persists with time through the model simulations. For example, when the snow cover is artificially removed in experiment “ExpNoSnow”, does it grow rapidly again due to the (inconsistent) atmospheric forcing?

Response:

1. Design of numerical experiments

The reviewer describes concerns related to the design of our numerical experiments, which are also concerns of another two reviewers. We performed a substantial revision on the numerical experiments. Please see Lines 304–319 in the revised manuscript, or see the paragraph below for convenience:

To test the response of atmospheric circulation to TPSC, two ensemble experiments are performed, namely, positive anomalous TPSC experiments (**ExpPOS**) and negative anomalous TPSC experiments (**ExpNEG**). The initial lower boundary condition TPSC is modified based on the IMS snow cover analysis. The simultaneous composites of the daily TPSC from the IMS snow cover analysis were treated as ‘anomalies’ (Fig. R1-2a). The climatological TPSC plus half of the ‘anomalies’, as shown in Fig. R1-2a, obtain the initial forcing for ExpPOS, whereas the initial forcing for ExpNEG is obtained by the climatological TPSC minus half of the ‘anomalies’, as shown in Fig. R1-2a. Hence, the difference in TPSC between ExpPOS and ExpNEG are the ‘anomalies’ shown in Fig. R1-2b. Note that the snow condition in the numerical model is snow depth. Adopted from BATS (Dickinson et al. 1993), the snow cover fraction (f_{sno}) is parameterized as a nonlinear function of snow depth (h_{sno}):

$$f_{sno} = h_{sno} / (10z_{g,0} + h_{sno}),$$

where $z_{g,0}$ (= 0.01 m) is the ground roughness length. We transformed the f_{sno} to snow depth by using this parameterized function. The snow depth in grids of IMS analysis was interpolated into model grid. As a result, the initial f_{sno} between ExpPOS and ExpNEG is the ‘anomalies’ (Fig. R1-2b).

2. Persistence of snow anomaly signal

Another issue the reviewer raises is on the persistence of the snow cover forcing in the model simulations.

We investigated the persistence of the snow anomaly signals in the observations and numerical experiments. We calculated the TPSCI in the IMS analysis (*Black line* in Fig. R1-6) and numerical experiments (*Blue line* in Fig. R1-6). The TPSCI in the IMS analysis represents the percent difference of the snow-covered area over the TP between the positive and negative events. Please see the composite section in the Methods for details. The TPSCI in the numerical experiments is the difference between ExpPOS and ExpNEG.

The composite at a 0-day lag of the TPSCI in the IMS analysis is 24.0%. In addition, the difference between the TPSCI of the ExpPOS and ExpNEG at the first day of simulation is 24.5%, which is almost equal to that of the composite because the experimental design is based on the composite of TPSC. Then, the TPSCI anomaly in both the observational and numerical experiment composites decreases with lag days (Fig. R1-6). The results show that the observed snow anomaly signal persists up to approximately one week and shows a decreasing tendency.

The numerical experiments can reproduce both the persistence of the snow anomaly signal and the decreasing tendency.

We have incorporated the above discussion into the revised manuscript. Please see Lines 191–199 in the revised manuscript. Fig. R1-6 is also Fig. 4b in the revised manuscript.

Figure R1-6 | The TPSCI from observations and numerical experiments. The *x*-axis represents the number of days lagging the start of each event for the composites or the model initial date. The *black line* and *blue line* represent the reanalysis/analysis composites and numerical experiments, respectively. The *light blue shading* represents the range of the EAJSI or TPSCI between the 25th and 75th percentile of the numerical experiment ensembles. The unit is %.

Reference

Dickinson, R. E., Henderson-Sellers, A. & Kennedy, P. J. Biosphere-atmosphere Transfer Scheme (BATS) Version 1e as Coupled to the NCAR Community Climate Model. *NCAR Technical Note* NCAR/TN-387+STR, doi:10.5065/D67W6959 (1993)

The impact of snow initialisation on subseasonal forecasts (encompassing the medium-range time scale explored here with the WRF model) has been examined in a broader context in several recent papers with global models. I find the paper lacking those relevant references. In particular, a cold regional anomaly throughout the troposphere was also modelled in subseasonal forecasts during high snow depth conditions over Tibet by Senan et al. (2016), albeit in springtime.

Orsolini, Y.J., Senan, R., Balsamo, G., Doblas-Reyes, F., Vitart, D., Weisheimer, A., Carrasco, A., Benestad, R. (2013), Impact of snow initialization on sub-seasonal forecasts, Clim. Dyn., 41:1969-1982.

Senan, R., Orsolini, Y.J., Weisheimer A., Vitart, F., Balsamo, G., Stockdale, T., Dutra, E., Doblas-Reyes, F., D. Basang, Impact of springtime Himalayan-Tibetan Plateau snowpack on the onset of the Indian summer monsoon in coupled seasonal forecasts, Clim. Dyn., Vol. 47, Issue 9, pp 2709–2725, doi:10.1007/s00382-016-2993-y. (2016)

Jeong, J.H., H.W. Linderholm, S.-H. Woo, C. Folland, B.-M. Kim, S.-J. Kim and D. Chen (2013), Impact of snow initialization on subseasonal forecasts of surface air temperature for the cold season, J. Clim., 26:1956-1972.

Response:

We agree with the reviewer that the impact of snow initialization on subseasonal forecasts is important to our particular work. The related papers have been included as references in the Introduction. Please see Lines 46–47 in the revised manuscript.

Few papers have examined this kind of response at the medium-range time scale, hence the results are quite novel. While the paper has merit, it is should be more polished to be considered for publication in Nature. Many of the statements could be made more precise. I recommend a major revision of the paper, before it is in an acceptable form for publication.

OTHER MAJOR COMMENTS

L123: the model composite and the reanalyses in Fig 1 are not completely similar. They differ over the TP itself, and over the Indian subcontinent. Some clarification is needed.

Response:

Fig. 1 in the previous manuscript is now Fig. 2 and Fig. 3 in the revised manuscript. As discussed in the manuscript, the numerical experiments reproduce the dipole structure of the U300 anomalies in response to TPSC variability found in the reanalysis. However, there is still a slight difference between the reanalysis composites and the numerical experiments response signal over the TP and the Indian subcontinent. The intensity of the zonal wind response in the numerical experiments is not as strong as that in the reanalysis composites. The slight difference between the reanalysis and model experiments could be caused by many factors, including model biases, model lateral boundary forcing constraints, which limit the model response to

TPSC changes, and the limited effect of the initial change in snow cover in the models (only over TP) compared to the real cases (different everywhere). Despite these biases, the numerical experiments could reproduce the major patterns and evolution of the U300 anomalies.

We have incorporated the above clarification into the revised manuscript. Please see Lines 137–147 in the revised manuscript.

Figure 1 : More results should be integrated in the main body of the paper than the “snap-shot” 6-day lag shown in Figure 1. Some material from Figures S3 and S4, showing other lags, should be used to expand figure 1 in order to give the reader an overview of the subseasonal evolution.

Response:

We agree with the reviewer that more results should be integrated in the main body of the paper to give the reader an overview of the subseasonal evolution. In the revised paper, we show the results at lag=1–8 days. Please see Fig. 1 and Fig. 2 in the revised manuscript.

L143 : the negative jet index response at short lags (lag 0 or 1) is not discussed. Nor is the strong zonal wind signal at lag 0 (seen in Figure S3), where the jet is enhanced on the southern side of the TP. Do the authors consider this as part of the near-instantaneous snow-induced response (as visible in temperature) ?

Response:

The subseasonal oscillation is an inherent phenomenon in the atmosphere (Knutson et al. 1986; Hsu 1996). The negative jet index at short lags (lag 0 or 1) should belong to the remaining signals in the negative phase of the last cycle.

Yes. We consider the strong zonal wind signal at lag = 0 days over the southern side of the TP as part of the near-instantaneous snow-induced response. However, the composite is stronger than that in the numerical experiments. This is because of the unavoidable difference of snow cover evolution between the real-world and ideal numerical experiment. The snow cover anomalies start before lag = 0 days (Fig. R1-7). These snow cover anomalies induce an atmospheric response that occurs in the composite study. However, in the numerical experiment,

the snow cover anomalies are added in the initial time step. As a result, the zonal wind on the southern side of the TP at a 0-day lag is stronger than that in numerical experiment.

Figure R1-7 | The TPSCI from observations. The percentage of snow-covered areas over the TP (TPSCI; unit is %) in the composite for the observed snow cover data (IMS analysis; *blue line*). The negative (positive) values of the *x*-axis represent leading (lagging) with respect to the start day.

References

- Hsu, H.-H. Global View of the intraseasonal Oscillation during Northern Winter. *J. Clim.* **9**, 2386–2406, doi:10.1175/1520-0442(1996)009<2386:GVOTIO>2.0.CO;2 (1996).
- Knutson, T. R., Weickmann, K. M. & Kutzbach, J. E. Global-Scale Intraseasonal Oscillations of Outgoing Longwave Radiation and 250 mb Zonal Wind during Northern Hemisphere Summer. *Mon. Weather Rev.* **114**, 605–623, doi:10.1175/1520-0493(1986)114<0605:GSIOOO>2.0.CO;2 (1986).

L164: I don't understand some points in the discussion of the surface energy balance. In presence of a thick snowpack, one expects also a cool surface anomaly (decreased sensible heat flux) due to the insulating property of snow (a thermodynamical and not radiative property). That should add to the same-sign temperature anomaly from the short-wave deficit. Some clarification is needed.

Response:

Thanks for your suggestion. We think you mentioned the snow-thermodynamic effect. We have revised the related statement. Please see Lines 179–184 in the revised manuscript, or see the paragraph below for convenience:

“Apart from the snow-albedo effect, snow cover insulates the overlying air from the soil layer below (snow-thermodynamic effect). The snow-thermodynamic also reduces the sensible heat flux at land surface. In contrast, the latent heat flux response is low. The overall responses of surface energy to the increased/decreased TPSC lead to anomalous cooling/heating effects, due to the snow-albedo effect and snow-thermodynamic effect.”

MINOR COMMENTS

L72-L74: much of that is a repeat of what was mentioned above.

Response:

Thanks for your suggestion. We have revised this part to make it more concise.

L 210: why aren't the temperature composites from the reanalyses shown ?

Response:

We have added a figure to show the temperature composites. Please see Figure S6 in the revised Supplementary Material.

Figure S9 : It would be useful to have the climatological winds in background as contours.

Response:

We have revised the related figure. Please see Figure 5d in the revised manuscript.

We appreciate the reviewer's valuable and thoughtful comments. We have incorporated these comments into the revised manuscript. The following is our point-by-point response. The blue italics are reviewer's comments. Our responses are provided in black.

Reviewer #2 (Remarks to the Author):

Review of "Influence of the Tibetan Plateau snow cover on East Asian atmospheric circulation on medium-range time-scales" by Wenkai Li etc..

Recommendation: Major revision.

Overview:

This study investigated the response of the atmosphere to the subseasonal variability of the TPSC. The motivation of this study based on the extensively investigation and discussion of the response of atmospheric variability to the Tibetan Plateau (TP) snow cover (TPSC) at interannual to decadal time-scales, rather than at subseasonal time-scale. Results showed that subseasonal variability of the TPSC can serve as an indicator of East Asian atmospheric circulation on medium-range time-scales (approximately 3–7 days in advance). The motivation of this study is a good one and could be useful to the subseasonal prediction. I would recommend a major revision before acceptance.

Major comments:

1. *The conclusion of this study is "The subseasonal variability of the TPSC influences the upper-level regional atmospheric circulations, especially the East Asia upper-level westerly jet stream (EAJS), during wintertime. The variability of the TPSC modulates the land surface thermal conditions over the TP, which acts as an elevated cooling source in the middle troposphere during wintertime. The upper-level geopotential height is affected by the TPSC and then the zonal wind. As a result, the EAJS is influenced". These mechanisms have long been well known to the community and can not be regarded as a new concept. Therefore, the discussion on the mechanism has no new idea and lack of clearly illustration and elaboration in physical perspective.*

Response:

We agree with the reviewer that the snow-albedo effect is well known to the community. In this paper, we use this classic theory to explain a new relationship that occurs *at faster subseasonal time scales*. To our knowledge, few studies have focused on the rapid subseasonal variability of TPSC and its atmospheric effect. Our work may help draw attention to such variability and its influences.

To emphasize the innovation of this study, we have rephrased and reorganized the related paragraphs (Lines 40–53 and Lines 55–83 in the revised manuscript). They are summarized as follows.

1. We demonstrated that the subseasonal variability of TPSC, which has previously received little attention, is an important component of the total variability. Subseasonal variability occurs over almost the entire TP. Our study further found that the subseasonal component explains more than half of the non-seasonal variation in the daily TPSC. This suggests that the subseasonal variability of the TPSC is nonnegligible.

2. It is therefore valuable to further investigate the atmospheric effects caused by such rapid variability of TPSC. A better understanding of the atmospheric effects of TPSC at multiple time scales, including subseasonal time scales, allows us to understand all aspects of atmospheric variability. Our results suggest that the subseasonal variability of the TPSC rapidly influences the East Asia atmospheric circulation and may serve as a predictor of East Asia atmospheric circulation at subseasonal time scales (approximately one week in advance).

2. The experiment design, mechanism illustration and conclusions in this study, actually emphasize the snow radiation cooling effect, these are known well by reader. An effective analysis of persistence of the snow cover cooling effects is needed. In fact, snow cover as a signal for predicts summer precipitation over Eastern China has been used for long time, but there is distinctly uncertainty.

Response:

Thanks for your good suggestion. We have paid more attention to the analysis of the persistence of snow cover cooling effects in the revised manuscript.

We investigated the persistence of the snow anomaly signals in the observations and numerical experiments. We calculated the TPSCI in the IMS analysis (*Black line* in Fig. R2-1)

and numerical experiments (*Blue line* in Fig. R2-1). The TPSCI in the IMS analysis represents the percent difference of the snow-covered area over the TP between the positive and negative events. Please see the composite section in the Methods for details. The TPSCI in the numerical experiments is the difference between ExpPOS and ExpNEG.

The composite at a 0-day lag of the TPSCI in the IMS analysis is 24.0%. In addition, the difference between the TPSCI of the ExpPOS and ExpNEG at the first day of simulation is 24.5%, which is almost equal to that of the composite because the experimental design is based on the composite of TPSC. Then, the TPSCI anomaly in both the observational and numerical experiment composites decreases with lag days (Fig. R2-1). The results show that the observed snow anomaly signal persists up to approximately one week and shows a decreasing tendency. The numerical experiments can reproduce both the persistence of the snow anomaly signal and the decreasing tendency.

We have incorporated the above discussion into the revised manuscript. Please see Lines 191–199 in the revised manuscript. Fig. R2-1 is also Fig. 4b in the revised manuscript.

Figure R2-1 | The TPSCI from observations and numerical experiments. The *x*-axis represents the number of days lagging the start of each event for the composites or the model initial date. The *black line* and *blue line* represent the reanalysis/analysis composites and numerical experiments, respectively. The *light blue shading* represents the range of the EAJSI or TPSCI between the 25th and 75th percentile of the numerical experiment ensembles. The unit is %.

The above results demonstrate that the numerical experiments can reproduce both the persistence of the snow anomaly signal and the decreasing tendency. A persistent surface energy response with a decreasing tendency is expected. Here, we take the SH as a representative component of land surface energy to show the persistence of the land surface energy response (Fig. R2-2). A strong negative response of sensible heat flux is caused by the snow-albedo effect

for the first day, as discussed above. Then, the sensible heat flux increases (absolute value decreases) with lag days. At a 6-day lag, the sensible heat flux response is -22.8 W/m^2 , implying that the surface energy can persist to a lag of approximately one week.

We have incorporated the above discussion into the revised manuscript. Please see Lines 199–206 in the revised manuscript. Fig. R2-2 is also Fig. S5 in the revised Supplementary.

Figure R2-2 | The response of the regionally averaged sensible heat flux (SH) over the TP surface to the subseasonal variability of TPSC in the numerical experiments. The unit is W/m^2 . The value is the difference between the SH of ExpPOS and ExpNEG. The x-axis represents the number of lag days from the initial date of the model.

3. Lack of rigorous verification with observations, increase/decrease albedo implies snow cover is uniform over TP, however, most of TP has no snow cover, even in wintertime. I am concerned about the experiment design and the possible mechanism that the authors did not fully addressed.

Response:

Many thanks for this valuable comment.

1. Characters of the subseasonal variation of the wintertime TPSC

We agree with the reviewer that most of the TP typically has no snow cover in wintertime. As shown in Fig. R2-3, most of the TP is covered by snow for 10–30% of wintertime. This suggests that most of the TP area has no snow cover for more than half of the wintertime period. However, for this reason, the subseasonal variability of the TPSC is significant during wintertime.

To conform confirm this, we analysed the standard deviation of the TPSC during wintertime by using daily snow cover data (Fig. R2-3a). Most areas have standard deviation greater than 20%. Meanwhile, some areas have values greater than 28% over the central and eastern TP (CTP and ETP). We also note that some parts of the western TP (marked by black oval in Fig. R2-3a) and the Himalayas have smaller standard deviations than areas over the CTP and ETP. The western TP and Himalayas are covered by snow during more than 90% of wintertime (Fig. R2-3b). The high climatological mean of the wintertime snow cover in these areas leads to smaller standard deviations because they are covered by snow almost all winter; however, these areas only represent a small percentage of the whole TP. For regions across almost the entire TP, especially the CTP and ETP, strong subseasonal variability generally occurs.

Figure 2-3 | Standard deviation and climatology of wintertime TPSC based on daily data. (a) Stand deviation of the 120-day high-pass filtered anomalous probability of snow-cover occurrence over the TP during wintertime. (b) Climatological probability of snow-cover occurrence over the TP during wintertime. The unit is %. Grey areas indicate areas with altitudes less than 3,000 metres.

We further selected a representative case to illustrate rapid snow cover changes (Fig. R2-4 and Fig. R2-5). Fig. R2-4 shows the TPSCI (*red line*) and its subseasonal anomalies (*blue line*) during the cold season of 2012/2013, while Fig. R2-5 shows the spatial snow cover on selected dates.

Although the climatology of the TPSCI (i.e., annual cycle) increases during this period (*grey line*), the TPSCI shows both increasing and decreasing changes in this season. For example, the TPSCI is 53.1% on 2012-12-29 (Fig. R2-5a), which is higher than its climatology. Then, the TPSCI decreases over the following 7 days. The TPSCI decreases to 24.2% on 2013-01-06 (Fig. 1-4b). Most of the TP area is without snow cover, except for some parts of the

western TP and the Himalayas. Then, the TPSCI increases over the month and reaches 63.9% on 2013-01-20 (Fig. 1-4c). Subsequently, the TPSCI decreases again and is close to its climatology on 2013-01-27 (Fig. 1-4d).

Figure R2-4 | The TPSCI and its subseasonal component in the cold season of 2012/2013. The *red line* shows the raw TPSCI (percentage of snow-covered area over the TP). The *blue line* shows the subseasonal component of the TPSCI. The *grey line* shows the annual cycle of the TPSCI. The *thin black line* shows the reference line of zero. The unit is %. (a)–(d) mark the selected dates on which the TPSC spatial distributions are shown in Fig. R2-5.

Figure R2-5 | The spatial distribution of TPSC on selected dates. The white area shows the snow-covered grids from the IMS snow cover analysis, while the orange area shows the grids with no snow. The right title in each subfigure represents the percentage of snow-covered area over the TP.

During the period from 2013-01-06 to 2013-01-27 (about half a month), the change in the TPSCI was up to 39.7%. The obvious subseasonal variation, such as the 39.7% change in the snow-covered area (approximately 1.1×10^6 km²) encompasses a broad area. The variability of snow cover over such a broad area may induce the atmospheric variability at a subseasonal time scale.

We have incorporated the above discussion into the revised manuscript and supplementary. Please see Lines 58–66 in the revised manuscript, and Fig. R2-3 is also Fig. 1. Please also see Lines 28–54 in the revised supplementary, and Fig. R2-4 and Fig. R2-5 are also Fig. S2 and Fig. S3.

2. Design of numerical experiments

The reviewer raises concerns about the design of our numerical experiments, which were also raised by another two reviewers. We did a substantial revision on the numerical experiments. Please see Lines 304–319 in the revised manuscript, or see the paragraph below for convenience:

To test the response of atmospheric circulation to TPSC, two ensemble experiments are performed, namely, positive anomalous TPSC experiments (**ExpPOS**) and negative anomalous TPSC experiments (**ExpNEG**). The initial lower boundary condition TPSC is modified based on the IMS snow cover analysis. The simultaneous composites of the daily TPSC from the IMS snow cover analysis were treated as ‘anomalies’ (Fig. R2-6a). The climatological TPSC plus half of the ‘anomalies’, as shown in Fig. R2-6a, obtain the initial forcing for ExpPOS, whereas the initial forcing for ExpNEG is obtained by the climatological TPSC minus half of the ‘anomalies’, as shown in Fig. R2-6a. Hence, the difference in TPSC between ExpPOS and ExpNEG are the ‘anomalies’ shown in Fig. R2-6b. Note that the snow condition in the numerical model is snow depth. Adopted from BATS (Dickinson et al. 1993), the snow cover fraction (f_{sno}) is parameterized as a nonlinear function of snow depth (h_{sno}):

$$f_{sno} = h_{sno} / (10z_{g,0} + h_{sno}),$$

where $z_{g,0}$ (= 0.01 m) is the ground roughness length. We transformed the f_{sno} to snow depth by using this parameterized function. The snow depth in grids of IMS analysis was interpolated into model grid. As a result, the initial f_{sno} between ExpPOS and ExpNEG is the ‘anomalies’ (Fig. R2-6b).

Figure R2-6 | The initial lower boundary condition TPSC based on the IMS snow cover analysis. (a) The simultaneous composites of the daily anomalous TP snow-cover probabilities for the IMS snow cover analysis. (b) The difference between the initial snow cover fraction of ExpPOS and ExpNEG. The unit is %. Grey areas indicate the areas with altitudes less than 3,000 metres.

3. Possible mechanism

We use the snow-albedo effect, which is a classical theory, to explain a newfound relationship that occurs at faster subseasonal time scales. The innovation of this study is to bring attention to such rapid short-term variability of the TPSC at the faster subseasonal time scales and the associated atmospheric effect. As mentioned in the response to your Comment #1, we have rephrased and reorganized the related paragraphs (Lines 40–53 and Lines 55–83 in the revised manuscript) to emphasize the innovation of this study,

Reference

Dickinson, R. E., Henderson-Sellers, A. & Kennedy, P. J. Biosphere-atmosphere Transfer Scheme (BATS) Version 1e as Coupled to the NCAR Community Climate Model. *NCAR Technical Note* NCAR/TN-387+STR, doi:10.5065/D67W6959 (1993)

4. Similar to comment 3, the snow data used in this study comes from the Interactive Multi-Sensor Snow and Ice Mapping System (IMS). The reliability of this dataset is not clear, which could result in an unclear or even a wrong conclusion.

Response:

Many thanks for this valuable comment. The reliability of the snow cover data is important for our study. This issue was also noted by Reviewer #3. Please see Lines 71–157 in the revised Supplementary, or see the paragraph below for convenience:

The IMS snow cover analysis is derived from a variety of data products, including satellite imagery and in situ data (Helfrich et al. 2007). Chen et al. (2012) validated the IMS snow cover analysis by a comparison with ground-based measurements over the continental United States. They found that the IMS maps demonstrate a good correspondence with the ground-based measurements. The daily rate of agreement between the products mostly ranges between 80% and 90% in the Northern Hemisphere during the winter season, when about a quarter to one third of the continental US territory is covered with snow. Furthermore, they suggested that, when mapping snow cover, IMS analysts use the same technique and similar sources of data (e.g., satellite imagery, in situ data, and automated snow remote sensing products) over the whole Northern Hemisphere. Therefore, it is reasonable to assume that the accuracy of snow cover mapping over the mid-latitude region of Eurasia is similar to that over North America. Yang et al. (2015) found that the overall accuracy of IMS snow cover analysis is higher than 91% compared to station observations over the TP. Based on Chen et al. (2012) and Yang et al. (2015), the IMS snow cover analysis should be reliable to measure the overall variability of the TPSC. To test this hypothesis, we also evaluated the reliability of the IMS snow cover analysis over the TP. Here, we evaluated not only its overall accuracy but also the subseasonal variability of the TPSC in the IMS dataset.

1. Station data

Figure R2-7 | Distribution of meteorological stations over the Tibetan Plateau (black dots). The purple contour marks the regions of the Tibetan Plateau with altitudes higher than 3,000 meters.

To assess the quality of the IMS snow cover analysis over the TP, we used daily snow depth data collected from national meteorological stations of the China Meteorological Administration (CMA). The station data cover the period from 2000–2010, and we studied 55 stations over the TP region (Fig. R2-7). The stations are mainly spread over the eastern TP. In

the central and western TP, the weather observations are particularly lacking due to the bitter natural geographical environment and sparse population. To our knowledge, these station data are the only long-term operational observations including snow cover over the TP. It should be noted that CMA meteorological stations tend to be located in inhabited valleys. This may lead to differences between the station observations and satellite observations.

2. Overall accuracy

We evaluated the overall accuracy (OA) of the IMS snow cover analysis over the TP. We used an evaluation method similar that of Chen et al. (2012) and Yang et al. (2015). The IMS pixel values (snow or no snow) were compared to the snow depth values of the station data. For a given station location that coincides with an IMS pixel, a ‘match’ was tagged if the IMS pixel value was ‘snow’ and the snow depth value from station data was ‘trace’ or greater. Similarly, if the IMS pixel was classified as ‘land’ and the station data recorded a snow depth of ‘0’, a ‘match’ occurred. A ‘mismatch’ occurred if either the IMS pixel was ‘snow’ and the station data showed a snow depth of ‘0’ or the IMS pixel was classified as ‘land’ and the station data showed a ‘trace’ or greater snow depth. If the station value was missing for that day, it was tagged as ‘missing’. Comparisons to the IMS data were not performed for those stations tagged as a ‘missing’. The daily agreement between the IMS snow pixels and station data was calculated as the number of matching pixels divided by the sum of matching pixels and mismatching pixels multiplied by 100%:

$$OA = \text{Matching pixels} / (\text{Matching pixels} + \text{Mismatching pixels}) \times 100\%.$$

The OA was calculated for each day during the period of 2000–2010 (sample N = 3650). The OA over the TP based on our validation evaluation is 92%. Yang et al. (2015) found that the OA of IMS snow cover analysis is higher than 91% over the TP. Our evaluation is consistent with that of Yang et al. (2015) Both our validation and Ref. 3 suggest that the OA of IMS snow cover analysis over the TP demonstrate a good correspondence with ground-based measurements.

3. Subseasonal variability

Our study focuses on the subseasonal variability of the TPSC. It is necessary to evaluate whether the IMS snow cover analysis can capture the subseasonal variability of the TPSC. Thus, the ‘TPSCI’ of both the station observations and IMS analysis for each winter during the period

of 2000–2010 are compared. Note that here, the ‘TPSCI’ is not the same as that in the manuscript. Here, the ‘TPSCI’ only covers the 55 stations shown in Fig. R2-7.

Also note that the snow condition observed at the stations is snow depth. Adopted from BATS (Dickinson et al. 1993), the snow cover fraction (f_{sno}) is parameterized as a nonlinear function of snow depth (h_{sno}):

$$f_{sno} = h_{sno}/(10z_{g,0}-h_{sno}),$$

where $z_{g,0}$ (= 0.01 m) is the ground roughness length. We estimate f_{sno} for each station by using this parameterized function. The average of f_{sno} for all 55 stations is the “TPSCI” of station observations.

The capture of the f_{sno} for one station by IMS analysis is estimated using the following steps. First, the five IMS snow cover grids nearest to one station are selected. Then, the average of these five grids is regarded as the capture of one station by IMS analysis. Finally, the average of the f_{sno} by IMS analysis for all 55 stations is regarded as the ‘TPSCI’ for IMS analysis. Similar to the method in the main text, we apply a 120-day high-pass filter to remove the interannual and decadal components for these two ‘TPSCI’ time series and focus on the subseasonal time scale.

The ‘TPSCI’ of both stations and IMS for each winter during the period of 2000–2010 are compared (Fig. R2-8). Overall, the IMS data capture the peaks of the ‘TPSCI’ anomaly well for each winter during the period of 2000–2010. For example, the above-anomaly to below-anomaly transitions are captured correctly by IMS analysis (Fig. R2-8i and Fig. R2-8j). The correlation coefficient (CC) between the two time series for each winter (the sample size $N = 120$) ranges from 0.39 to 0.79. The CC for the two time series for all 10 winters is 0.56 (the sample size $N = 1200$), indicating that the IMS snow cover analysis captures the subseasonal variability of the TPSC well.

The above evaluation shows that the IMS snow cover analysis can capture the general subseasonal variability of the TPSC. This dataset is valuable, especially for the TP, where observations are particularly lacking due to the bitter natural geographical environment and sparse population.

Figure R2-8 | The subseasonal variability of the TPSCI from stations and IMS snow cover analysis. Daily time evolution of the TPSCI for 55 stations (*red lines*) and IMS snow cover analysis (*blue lines*). The time series for each winter are standardized. The right title in each plot shows the correlation coefficient (CC) between the two time series for each winter (the sample size $N = 120$). The CC for the two time series for all 10 winters is 0.56 (the sample size $N = 1200$).

References

- Helfrich, S. R., McNamara, D., Ramsay, B. H., Baldwin, T. & Kasheta, T. Enhancements to, and forthcoming developments in the Interactive Multisensor Snow and Ice Mapping System (IMS). *Hydrol. Process.* **21**, 1576–1586, doi:10.1002/hyp.6720 (2007).
- Chen, C., Lakhankar, T., Romanov, P., Helfrich, S., Powell, A. & Khanbilvardi, R. Validation of NOAA-Interactive Multisensor Snow and Ice Mapping System (IMS) by Comparison with Ground-Based Measurements over Continental United States. *Remote Sens.* **4**, 1134–1145, doi:10.3390/rs4051134 (2012).
- Yang, J. *et al.* Evaluation of snow products over the Tibetan Plateau. *Hydrol. Process.* **29**, 3247–3260, doi:10.1002/hyp.10427 (2015).
- Dickinson, R. E., Henderson-Sellers, A. & Kennedy, P. J. Biosphere-atmosphere Transfer Scheme (BATS) Version 1e as Coupled to the NCAR Community Climate Model. *NCAR Technical Note NCAR/TN-387+STR*, doi:10.5065/D67W6959 (1993).

5. Uncertainty of the definition of TPSCI. TPSCI is defined with the IMS analysis data, As mentioned in comment 4, the datasets may have great uncertainties in representing the real snow cover distributions, thus, it is not clear whether the definition of TPSCI is reasonable or not, especially in the western TP.

Response:

Many thanks for this comment. The TPSCI here represents the percentage of snow-covered area over the TP. As shown in the response to your Comment #3, the subseasonal variability occurs over almost the entire TP (Fig. R2-3a), except for some parts of the western TP. We focus on the subseasonal variation of TPSC over the entire TP. The snow index is constructed using a broad longitudinal extent covering the entire TP.

To confirm whether the TPSCI can represent the overall character of subseasonal variations of the TPSC, we performed a composite analysis for TPSC (Fig. R2-6a). The composite method here is the same as the method we used in our manuscript. Please see Methods in the revised manuscript for details on the composite analysis. Most TP areas have a composite value higher than 20%. There are even high composite values greater than 30% and sometimes greater than 40% that occur over the CTP and ETP. Generally, TPSCI variations encompass almost the entire TP. This result is consistent with the total subseasonal variability

(Fig. R2-3a), suggesting that the TPSCI can represent the overall character of subseasonal variations of TPSC.

6. Potential errors and misleadings of the model configuration and experiment design. First, WRF model is a collaborative achievement among NCAR, NCEP, ESRL, AFWA and NRL etc.; Second, the authors conducted the sensitivity experiments by roughly increasing or removing snow over the whole TP. Since the snow over the TP has great differences in spatial-temporal, previous studies have revealed and highlighted that snow cover in different region has much different impacts on the East Asian atmospheric circulation. Therefore, the experiment design is not reasonable and the corresponding results are not convincing to this reviewer.

Response:

The reviewer is concerned about the design of our numerical experiments, which is also a concern of Reviewer #2 and Reviewer #3. We performed substantial revisions on the numerical experiments. Please refer to the response to your Comment #3.

We agree with the reviewer that snow cover in different regions have different impacts on the East Asian atmospheric circulation. In the revised numerical experiment, the TPSC in the initial lower boundary condition is modified based on the IMS snow cover analysis. The difference in the spatial distribution between the initial snow of ExpPOS and ExpNEG is derived from the IMS composite (Fig. R2-6b). The snow cover forcing mainly extends between the Kunlun Mountains and Tanggula Mountains over the central and eastern TP. Thus, the spatial distribution of snow cover is considered in the revised numerical experimental design.

7. The color bar in Fig. 1 is not consistent in each panel, which is misleading to this reviewer because the intensity and structure of regional upper-level zonal wind are not well reflected in the simulation.

Response:

Many thanks for this valuable comment. As discussed in the manuscript, the numerical experiment reproduce the dipole structure of the U300 anomalies in response to TPSC variability found in the reanalysis. But the intensity of the zonal wind response in the numerical

experiments is not as strong as that in the reanalysis composites, so we used different coloured bars in these two figures. The difference between the reanalysis and model experiments could be caused by many factors, including model biases, model lateral boundary forcing constraints, which limit the model response to TPSC changes, and the limited effect of the initial change in snow cover in the models (only over TP) compared to the real cases (different everywhere). Despite these biases, the numerical experiments could reproduce the major patterns and evolution of the U300 anomalies.

We have incorporated the above clarification into the revised manuscript. Please see Lines 137–147 in the revised manuscript.

We appreciate the reviewer's valuable and thoughtful comments. We have incorporated these comments into the revised manuscript. The following is our point-by-point response. The blue italics are reviewer's comments. Our responses are provided in black.

Reviewer #3 (Remarks to the Author):

Influence of the Tibetan Plateau snow cover on East Asian atmospheric circulation on medium-range time-scales

By Li et al.

Recommendation: Major revisions

General Comments:

This study examines Influence of the Tibetan Plateau (TP) snow cover on East Asian atmospheric circulation on medium-range time-scale, which has been noticed by few previous works. They proposed the physical linkage is the initial surface sensible heating reduce, not surprise. The results are reasonable but need to clarify several issues. Therefore, I would recommend major revisions.

1. For the snow data used in this study. If this data is valid on daily time scale as well as over the TP? You need to show the evidence.

Response:

Many thanks for this valuable comment. The reliability of the snow cover data is important for our study. This issue was also noted by Reviewer #2. Please see Lines 71–157 in the revised Supplementary, or see the paragraph below for convenience:

The IMS snow cover analysis is derived from a variety of data products, including satellite imagery and in situ data (Helfrich et al. 2007). Chen et al. (2012) validated the IMS snow cover analysis by a comparison with ground-based measurements over the continental United States. They found that the IMS maps demonstrate a good correspondence with the ground-based measurements. The daily rate of agreement between the products mostly ranges between 80% and 90% in the Northern Hemisphere during the winter season, when about a quarter to one third of the continental US territory is covered with snow. Furthermore, they suggested that,

when mapping snow cover, IMS analysts use the same technique and similar sources of data (e.g., satellite imagery, in situ data, and automated snow remote sensing products) over the whole Northern Hemisphere. Therefore, it is reasonable to assume that the accuracy of snow cover mapping over the mid-latitude region of Eurasia is similar to that over North America. Yang et al. (2015) found that the overall accuracy of IMS snow cover analysis is higher than 91% compared to station observations over the TP. Based on Chen et al. (2012) and Yang et al. (2015), the IMS snow cover analysis should be reliable to measure the overall variability of the TPSC. To test this hypothesis, we also evaluated the reliability of the IMS snow cover analysis over the TP. Here, we evaluated not only its overall accuracy but also the subseasonal variability of the TPSC in the IMS dataset.

1. Station data

To assess the quality of the IMS snow cover analysis over the TP, we used daily snow depth data collected from national meteorological stations of the China Meteorological Administration (CMA). The station data cover the period from 2000–2010, and we studied 55 stations over the TP region (Fig. R3-1). The stations are mainly spread over the eastern TP. In the central and western TP, the weather observations are particularly lacking due to the bitter natural geographical environment and sparse population. To our knowledge, these station data are the only long-term operational observations including snow cover over the TP. It should be noted that CMA meteorological stations tend to be located in inhabited valleys. This may lead to differences between the station observations and satellite observations.

Figure R3-1 | Distribution of meteorological stations over the Tibetan Plateau (black dots). The purple contour marks the regions of the Tibetan Plateau with altitudes higher than 3,000 meters.

2. Overall accuracy

We evaluated the overall accuracy (OA) of the IMS snow cover analysis over the TP. We used an evaluation method similar that of Chen et al. (2012) and Yang et al. (2015). The IMS pixel values (snow or no snow) were compared to the snow depth values of the station data. For a given station location that coincides with an IMS pixel, a ‘match’ was tagged if the IMS pixel value was ‘snow’ and the snow depth value from station data was ‘trace’ or greater. Similarly, if the IMS pixel was classified as ‘land’ and the station data recorded a snow depth of ‘0’, a ‘match’ occurred. A ‘mismatch’ occurred if either the IMS pixel was ‘snow’ and the station data showed a snow depth of ‘0’ or the IMS pixel was classified as ‘land’ and the station data showed a ‘trace’ or greater snow depth. If the station value was missing for that day, it was tagged as ‘missing’. Comparisons to the IMS data were not performed for those stations tagged as a ‘missing’. The daily agreement between the IMS snow pixels and station data was calculated as the number of matching pixels divided by the sum of matching pixels and mismatching pixels multiplied by 100%:

$$OA = \text{Matching pixels} / (\text{Matching pixels} + \text{Mismatching pixels}) \times 100\%.$$

The OA was calculated for each day during the period of 2000–2010 (sample N = 3650). The OA over the TP based on our validation evaluation is 92%. Yang et al. (2015) found that the OA of IMS snow cover analysis is higher than 91% over the TP. Our evaluation is consistent with that of Yang et al. (2015) Both our validation and Ref. 3 suggest that the OA of IMS snow cover analysis over the TP demonstrate a good correspondence with ground-based measurements.

3. Subseasonal variability

Our study focuses on the subseasonal variability of the TPSC. It is necessary to evaluate whether the IMS snow cover analysis can capture the subseasonal variability of the TPSC. Thus, the ‘TPSCI’ of both the station observations and IMS analysis for each winter during the period of 2000–2010 are compared. Note that here, the ‘TPSCI’ is not the same as that in the manuscript. Here, the ‘TPSCI’ only covers the 55 stations shown in Fig. R3-1.

Also note that the snow condition observed at the stations is snow depth. Adopted from BATS (Dickinson et al. 1993), the snow cover fraction (f_{sno}) is parameterized as a nonlinear function of snow depth (h_{sno}):

$$f_{sno} = h_{sno} / (10z_{g,0} - h_{sno}),$$

where $z_{g,0}$ (= 0.01 m) is the ground roughness length. We estimate f_{sno} for each station by using this parameterized function. The average of f_{sno} for all 55 stations is the “TPSCI” of station observations.

The capture of the f_{sno} for one station by IMS analysis is estimated using the following steps. First, the five IMS snow cover grids nearest to one station are selected. Then, the average of these five grids is regarded as the capture of one station by IMS analysis. Finally, the average of the f_{sno} by IMS analysis for all 55 stations is regarded as the ‘TPSCI’ for IMS analysis. Similar to the method in the main text, we apply a 120-day high-pass filter to remove the interannual and decadal components for these two ‘TPSCI’ time series and focus on the subseasonal time scale.

The ‘TPSCI’ of both stations and IMS for each winter during the period of 2000–2010 are compared (Fig. R3-2). Overall, the IMS data capture the peaks of the ‘TPSCI’ anomaly well for each winter during the period of 2000–2010. For example, the above-anomaly to below-anomaly transitions are captured correctly by IMS analysis (Fig. R3-2i and Fig. R3-2j). The correlation coefficient (CC) between the two time series for each winter (the sample size $N = 120$) ranges from 0.39 to 0.79. The CC for the two time series for all 10 winters is 0.56 (the sample size $N = 1200$), indicating that the IMS snow cover analysis captures the subseasonal variability of the TPSC well.

The above evaluation shows that the IMS snow cover analysis can capture the general subseasonal variability of the TPSC. This dataset is valuable, especially for the TP, where observations are particularly lacking due to the bitter natural geographical environment and sparse population.

References

- Helfrich, S. R., McNamara, D., Ramsay, B. H., Baldwin, T. & Kasheta, T. Enhancements to, and forthcoming developments in the Interactive Multisensor Snow and Ice Mapping System (IMS). *Hydrol. Process.* **21**, 1576–1586, doi:10.1002/hyp.6720 (2007).
- Chen, C., Lakhankar, T., Romanov, P., Helfrich, S., Powell, A. & Khanbilvardi, R. Validation of NOAA-Interactive Multisensor Snow and Ice Mapping System (IMS) by Comparison with Ground-Based Measurements over Continental United States. *Remote Sens.* **4**, 1134–1145, doi:10.3390/rs4051134 (2012).
- Yang, J. *et al.* Evaluation of snow products over the Tibetan Plateau. *Hydrol. Process.* **29**, 3247–3260, doi:10.1002/hyp.10427 (2015).

Dickinson, R. E., Henderson-Sellers, A. & Kennedy, P. J. Biosphere-atmosphere Transfer Scheme (BATS) Version 1e as Coupled to the NCAR Community Climate Model. *NCAR Technical Note NCAR/TN-387+STR*, doi:10.5065/D67W6959 (1993).

Figure R3-2 | The subseasonal variability of the TPSCI from stations and IMS snow cover analysis. Daily time evolution of the TPSCI for 55 stations (*red lines*) and IMS snow cover analysis (*blue lines*). The time series for each winter are standardized. The right title in each plot shows the correlation coefficient (CC) between the two time series for each winter (the sample size $N = 120$). The CC for the two time series for all 10 winters is 0.56 (the sample size $N = 1200$).

2. How the snow anomaly signal maintains in the observation and ExpSnow? How the surface energy change up to Lag=6?

Response:

Many thanks for your comments. This issue is also raised by Reviewer #2.

1. Snow anomaly signal

We investigated the persistence of the snow anomaly signals in the observations and numerical experiments. We calculated the TPSCI in the IMS analysis (*Black line* in Fig. R3-3) and numerical experiments (*Blue line* in Fig. R3-3). The TPSCI in the IMS analysis represents the percent difference of the snow-covered area over the TP between the positive and negative events. Please see the composite section in the Methods for details. The TPSCI in the numerical experiments is the difference between ExpPOS and ExpNEG.

The composite at a 0-day lag of the TPSCI in the IMS analysis is 24.0%. In addition, the difference between the TPSCI of the ExpPOS and ExpNEG at the first day of simulation is 24.5%, which is almost equal to that of the composite because the experimental design is based on the composite of TPSC. Then, the TPSCI anomaly in both the observational and numerical experiment composites decreases with lag days (Fig. R3-3). The results show that the observed snow anomaly signal persists up to approximately one week and shows a decreasing tendency. The numerical experiments can reproduce both the persistence of the snow anomaly signal and the decreasing tendency.

We have incorporated the above discussion into the revised manuscript. Please see Lines 191–199 in the revised manuscript. Fig. R3-3 is also Fig. 4b in the revised manuscript.

Figure R3-3 | The TPSCI from observations and numerical experiments. The x-axis represents the number of days lagging the start of each event for the composites or the model initial date. The *black line* and *blue line*

represent the reanalysis/analysis composites and numerical experiments, respectively. The *light blue shading* represents the range of the EAJSI or TPSCI between the 25th and 75th percentile of the numerical experiment ensembles. The unit is %.

2. Land surface energy

The above results demonstrate that the numerical experiments can reproduce both the persistence of the snow anomaly signal and the decreasing tendency. A persistent surface energy response with a decreasing tendency is expected. Here, we take the SH as a representative component of land surface energy to show the persistence of the land surface energy response (Fig. R3-4). A strong negative response of sensible heat flux is caused by the snow-albedo effect for the first day, as discussed above. Then, the sensible heat flux increases (absolute value decreases) with lag days. At a 6-day lag, the sensible heat flux response is -22.8 W/m^2 , implying that the surface energy can persist to a lag of approximately one week.

We have incorporated the above discussion into the revised manuscript. Please see Lines 199–206 in the revised manuscript. Fig. R3-4 is also Fig. S5 in the revised Supplementary.

Figure R3-4 | The response of the regionally averaged sensible heat flux (SH) over the TP surface to the subseasonal variability of TPSC in the numerical experiments. The unit is W/m^2 . The value is the difference between the SH of ExpPOS and ExpNEG. The x-axis represents the number of lag days from the initial date of the model.

3. *For the physics of TP snow impact, what are the differences between medium-range and interannual time-scales? What's the new insight?*

Response:

Thanks for your comment. The snow cover influences climate through physics, mainly via the snow-albedo effect, snow-hydrology effect and snow-thermodynamic effect. Studies on the climate effects of snow cover at early (e.g., Barnett et al. 1988; Yasunari et al. 1991; Cohen and Rind 1991) to late stages (e.g., Xian and Duan 2016; Gastineau et al. 2017; Wang et al. 2018) involve these physics. In this paper, we further use this classical theory to explain a newfound relationship that occurs *at shorter subseasonal time scales*.

The new insights in our manuscript reveal the rapid subseasonal variability of TPSC and the associated atmospheric effects, which have largely been ignored before. Our study found that the subseasonal component explains more than half of the non-seasonal variations in daily TPSC. This suggests that the subseasonal variability of the TPSC is nonnegligible.

A better understanding of the atmospheric effects of TPSC at multiple time scales, including subseasonal time scales, allows us to understand all aspects of atmospheric variability. Hence, it is valuable to further investigate the atmospheric effects caused by this rapid component of variability. Our results suggest that the subseasonal variability of the TPSC rapidly influences the East Asia atmospheric circulation at shorter time scales. TPSC may serve as an indicator of East Asia atmospheric circulation at medium-range time scales (approximately one week in advance).

To our knowledge, few studies have focused on the rapid subseasonal variability of TPSC. Our work may help bring attention to the short-term variability of TPSC. More discussion on the rapid subseasonal variability of TPSC was added to emphasize the highlights of our work. we have rephrased and reorganized the related paragraphs (Lines 40–53 and Lines 55–83 in the revised manuscript).

References

- Barnett, T. P., Dumenil, L., Schlese, U. & Roeckner, E. The effect of Eurasian snow cover on global climate. *Science* **239**, 504, doi:10.1126/science.239.4839.504 (1988).
- Yasunari, T., Kitoh, A. & Tokioka, T. Local and Remote Responses to Excessive Snow Mass over Eurasia Appearing in the Northern Spring and Summer Climate A Study with the MRI-GCM. *Meteorol. Soc. Jpn.* **69**, 473–487, doi:10.2151/jmsj1965.69.4_473 (1991).
- Cohen, J. & Rind, D. The Effect of Snow Cover on the Climate. *J. Clim.* **4**, 689–706, doi:10.1175/1520-0442(1991)004<0689:TEOSCO>2.0.CO;2 (1991).

- Xiao, Z. & Duan, A. Impacts of Tibetan Plateau Snow Cover on the Interannual Variability of the East Asian Summer Monsoon. *J. Clim.* **29**, 8495–8514, doi:10.1175/JCLI-D-16-0029.1 (2016).
- Gastineau, G., García-Serrano, J. & Frankignoul, C. The Influence of Autumnal Eurasian Snow Cover on Climate and Its Link with Arctic Sea Ice Cover. *J. Clim.* **30**, 7599–7619, doi:10.1175/JCLI-D-16-0623.1 (2017).
- Wang, Z. et al. Influence of Western Tibetan Plateau Summer Snow Cover on East Asian Summer Rainfall. *J. Geophys. Res. Atmos.* **123**, 2371–2386, doi:10.1002/2017JD028016 (2018).

4. *L30, Add 'during wintertime' after "in advance)". You didn't study other seasons.*

Response:

We have revised this statement according to your suggestion.

5. *L45, add 'at surface' before 'at the same lat' since it is not the case at pressure surface.*

Response:

We have revised this statement according to your suggestion.

6. *Fig. 1, the u300 anomaly is strong at lag=0, indicating the snow anomaly's contributor. Need discussion here.*

Response:

Thanks for your suggestion. This issue is also raised by Reviewer #1.

We consider the strong zonal wind signal at lag = 0 days over the southern side of the TP as part of the near-instantaneous snow-induced response. However, the composite is stronger than that in the numerical experiments. This is because of the unavoidable difference of snow cover evolution between the real-world and ideal numerical experiment. The snow cover anomalies start before lag = 0 days (Fig. R3-5). These snow cover anomalies induce an atmospheric response that occurs in the composite study. However, in the numerical experiment, the snow cover anomalies are added in the initial time step. As a result, the zonal wind on the southern side of the TP at a 0-day lag is stronger than that in numerical experiment.

Figure R3-5 | The TPSCI from observations. The percentage of snow-covered areas over the TP (TPSCI; unit is %) in the composite for the observed snow cover data (IMS analysis; *blue line*). The negative (positive) values of the *x*-axis represent leading (lagging) with respect to the start day.

7. L169, Table 1, observation and numerical simulations. Which one? Only one number in Table 1 for one variable.

Response:

Table 1 is for numerical simulations. We have revised this statement. Please see Line 185–190 in the revised manuscript.

8. L249, great than 3000 m. How about 2000 m or 2500 m. If the results are sensitive to the altitude for snow index?

Response:

We calculated the TPSCI by using IMS analysis with altitude greater than 2000 m, 2500 m and 3000 m. Then we perform same composite method by using these three TPSCIs, respectively (Fig. R3-6). As shown in Figure R3-6, the choice of the criterion of altitude have slight influence on the results. But the pattern and intensity of the composite keep almost the same. The results are not sensitive to the criterion of altitude to calculate the TPSCI.

Figure R3-6 | Sensitivity of the results to the criterion of the TPSCI. As Figure 1c in the revised manuscript, but these composite of U300 are with respect to the TPSCI calculated by using the snow cover data over the TP at an altitude of greater than (a) 2,000 metres, (b) 2,500 metres, (c) 3,000 metres. The purple contour outline marks the regions of the Tibetan Plateau with altitudes of greater than 2,000 metres, 2,500 metres and 3,000 metres.

Reviewers' Comments:

Reviewer #1:

Remarks to the Author:

The authors have done substantial revisions to the manuscript, which has gained in clarity. I commend the authors for their extensive revisions, which now include a comparison of IMS snow cover with in-situ station data. The authors have clarified many issues raised by reviewers, including in their numerical experiments.

I, however, think that the paper still needs some major revisions, before it is acceptable for publication in Nature.

A first minor comment is that the target of the paper (according to the title) is the subseasonal time scale, which falls between the short-to-medium time scale of weather prediction and the longer seasonal time scale. This subseasonal time scale is a major focus of current international (e.g. WCRP) research programmes. However, all the results shown cover a 8-day period at the maximum, so this is rather a study on the impact of the TP at the short-to-medium time scale, not truly at the subseasonal scale. I find the original title (in the submitted manuscript) was more accurate than the revised one. Some careful rewording is to be made. For example, L53, L238 or L235 "We extended the time scale of the prediction...". Also, this is more a sensitivity study since there is no verification of the "prediction".

I am somewhat concerned by the over-persistency of the snow signal in the model experiments. The authors state that (L199) the numerical experiments can reproduce both the persistence of the snow anomaly and its decreasing tendency. Rather, I am concerned by the over-persistency. The observations clearly reveal that the snow events are transient and short-lived (Fig S2 or Fig. 4b). In Fig. 4b or Fig. S5, it seems that the model snow anomaly persists well beyond the 8-day period. It would be worth to discuss the implications, since the model may over-represent the snow forcing. Over the Tibetan Plateau in winter, snowfall events and snow accumulation are shortened by the wind drift and snow sublimation, processes that might not be well represented by the model. A short discussion is needed here. Information on the evolution of the actual mean snow depth in the two model experiments could be incorporated in the Supplementary figures. I would think it is worth to show both experiments, as snow depth might grow rapidly again after initial removal due to the (inconsistent with removal) atmospheric forcing?

I had raised the issue of the causality of the relation between snow and jet stream variability. For example, one might argue that travelling upper-tropospheric disturbances embedded in the initial conditions could be responsible for both the snowfall and the downstream jet variability. There is – actually – an initial negative anomaly in the jet index (Fig. 4a). The model experiments do confirm the snow feedback upon the jet stream, yet, it is smaller than in the observations (Fig. 4a). Could it be that the snow exerts that feedback but that some of the jet index variability comes from the atmospheric initial conditions? Some comments are warranted.

OTHER MAJOR COMMENTS

Regarding how the land conditions are initialised, is it only the snow cover that is changed between the two experiments (i.e. what about the other land variables such as snow temperature, density, subsurface temperature and so forth)? This should be made clearer in the text.

In particular, I wonder if the IMS data be a slight overestimation of the snow cover. The authors use the 24km-resolution IMS binary data. I believe that if a 24km² grid box is snow covered (value 1) in IMS, it means that 50% of the area is snow covered, and it is not (value 0) if less than 50% of the grid box is covered. The original data has a pixel resolution of 4km. Although the agreement of IMS with station snow cover (after conversion from snow depth) in Fig S8 is convincing, a brief description of how the original IMS is handled, is warranted. In particular, I

wonder if the snow cover extent derived from IMS consistent with other studies, e.g., based on MODIS satellite snow cover? A sentence or two on other recent studies might be worth here.

MINOR COMMENTS

Figure S4 contains little information; it could be more instructive to show the climatology as background contours in Figure 2, to see how anomalies reinforce or weaken climatological winds.

L 109: "In contrast,..". This sentence is not necessary.

L131: remove "experiments."

L188: the 2nd "downward" should be "upward".

L201 : Has "SH" be defined in the text ? (or only in the Table)

L214: the cooling response. The reference to Senan et al. should be mentioned here since it is a similar finding.

L219-221: the word "adaptative" is not appropriate here. The word "affect" is used twice in the same sentence. Rather use : The zonal wind is in geostrophic balance with geopotential heights (?)

Caption of S9: "simultaneous composites of the daily anomalous TP snow-cover probabilities for the IMS snow cover analysis." That wording is quite unclear.

Reviewer #2:

Remarks to the Author:

Comment on "Influence of the Tibetan Plateau snow cover on East Asian atmospheric circulation on medium-range time-scales" by Li et al.

General comments

The revised manuscript has sufficiently addressed my concerns on its scientific content. I recommend it to be published after the following minor comments are addressed.

Minor comments

- 1.The mechanism linked to effects of TPSC variability on East Asian atmospheric circulation at subseasonal scale is still need described clearly.
- 2.The abstract should be further summarized to show the highlight in this research.
- 3.The uncertainties in results analysis should be discussed.

Reviewer #3:

None

We thank the Reviewers for their valuable and thoughtful comments that helped us clarify our methods and strengthen the scientific discussion in the revised manuscript.

The following is our point-by-point response. The blue italics are reviewers' comments. Our responses are provided in black. All line numbers refer to **the version of the manuscript and supplementary material with tracked changes**.

Reviewer #1 (Remarks to the Author):

The authors have done substantial revisions to the manuscript, which has gained in clarity. I commend the authors for their extensive revisions, which now include a comparison of IMS snow cover with in-situ station data. The authors have clarified many issues raised by reviewers, including in their numerical experiments.

I, however, think that the paper still needs some major revisions, before it is acceptable for publication in Nature.

A first minor comment is that the target of the paper (according to the title) is the subseasonal time scale, which falls between the short-to-medium time scale of weather prediction and the longer seasonal time scale. This subseasonal time scale is a major focus of current international (e.g. WCRP) research programmes. However, all the results shown cover a 8-day period at the maximum, so this is rather a study on the impact of the TP at the short-to-medium time scale, not truly at the subseasonal scale. I find the original title (in the submitted manuscript) was more accurate than the revised one. Some careful rewording is to be made. For example, L53, L238 or L235 “We extended the time scale of the prediction...”. Also, this is more a sensitivity study since there is no verification of the “prediction”.

Response:

1. Yes, we completely agree with the reviewer about the terminologies of the timescale. In this study, we found that the subseasonal TPSC exerts significant influences on the upper-level circulation at the timescale of 3–8 days. The *Glossary of Meteorology*, which is peer-reviewed and published by the American Meteorological Society, indicates that “medium-range” is a period extending from about three days to seven days in advance. Thus, we modified the title and related descriptions by using “medium-range” to describe the timescale of 3–8 days in advance in the revised manuscript.
2. We also agree that this study does not focus on prediction. Instead, we aimed to investigate the influences of subseasonal TPSC on the upper level atmospheric circulation over East Asia, which could provide some information for prediction. Therefore, we removed the descriptions about prediction in the revised manuscript.

References

American Meteorological Society, cited 2018: Medium-range forecast. *Glossary of Meteorology*. [Available online at [http://glossary.ametsoc.org/wiki/medium-range forecast](http://glossary.ametsoc.org/wiki/medium-range_forecast).]

I am somewhat concerned by the over-persistency of the snow signal in the model experiments. The authors state that (L199) the numerical experiments can reproduce both the persistence of the snow anomaly and its decreasing tendency. Rather, I am concerned by the over-persistency. The observations clearly reveal that the snow events are transient and short-lived (Fig S2 or Fig. 4b). In Fig. 4b or Fig. S5, it seems that the model snow anomaly persists well beyond the 8-day period. It would be worth to discuss the implications, since the model may over-represent the snow forcing. Over the Tibetan Plateau in winter, snowfall events and snow accumulation are shortened by the wind drift and snow sublimation, processes that might not be well represented by the model. A short discussion is needed here. Information on the evolution of the actual mean snow depth in the two model experiments could be incorporated in the Supplementary figures. I would think it is worth to show both experiments, as snow depth might grow rapidly again after initial removal due to the (inconsistent with removal) atmospheric forcing?

Response:

1. We thank and agree with the reviewer. The snow cover changes are complicated in reality. The numerical model, mainly its land surface scheme, might have shortcomings in completely reproducing the snow cover changes. For example, the Noah-MP land surface scheme does not include the process of wind-induced snow transport. For our numerical experiments, the model had some difficulties reproducing the sharp TPSC change over 7 days. However, the numerical experiments can reproduce both the persistence of the snow anomaly signal and the decreasing tendency within 6 days. The rapid influence of TPSC on the atmosphere is at a time scale of approximately 3–8 days. Since we focus on the medium-range (3–8 days) response, the bias occurring beyond 7 days would not change the current conclusions.

To honestly reflect the model biases, we added a statement (Lines 222–229) in the revised manuscript: “The numerical experiments can reproduce both the persistence of the snow anomaly signal and the decreasing tendency within 6 days. The observational TPSCI anomalies fall sharply after 7 days, while the model that simulated TPSCI anomalies still shows a relatively slow decreasing tendency. The persistences of the TPSCI in ExpPOS and ExpNEG show similar results (Supplementary Note 2 and Supplementary Fig. 4). Although there is some bias 7 days after the initial date, the numerical experiments can generally

reproduce both the persistence of the snow anomaly signal and the decreasing tendency at a medium range.”

Moreover, a discussion of model uncertainties was also added to Lines 286–289 of the revised manuscript: “Although the model reproduces both the persistence of the snow anomaly signal and its tendency at the medium range, the model has some biases in simulating the sharp variations in TPSCI after 7 days of observations. More works are necessary to further improve the snow and snow-atmosphere interaction processes in land surface models.”

2. Fig. R1-1 shows the persistence of the TPSCI in both the positive TPSC and negative TPSC composites/experiments. Fig. R1-1 is also Supplementary Fig. 4 in the revised supplementary material. An explanation for Supplementary Fig. 4 is presented in Lines 63–69 in the revised supplementary material (Supplementary Note 2).

Figure R1-1 | The persistence of the TPSCI. This figure is similar to Figure 4b in the revised manuscript, but show the results in both the positive TPSC and negative TPSC composites/experiments. The x-axis represents the number of days lagging the start of each event for the composites or the model initial date. The *dashed red line* and *dashed blue line* represent the analysis composites for positive and negative TPSC events, respectively. The *solid red line* and *solid blue line* represent numerical experiments of ExpPOS and ExpNEG, respectively. The *light shadings* represent the range of the TPSCI between the 25th and 75th percentile of the numerical experiment ensembles. The unit is %.

I had raised the issue of the causality of the relation between snow and jet stream variability. For example, one might argue that travelling upper-tropospheric disturbances embedded in the initial conditions could be responsible for both the snowfall and the downstream jet variability. There is –actually– an initial negative anomaly in the jet index (Fig. 4a). The model experiments do confirm the snow feedback upon the jet stream, yet, it is smaller than in the observations (Fig.

4a). Could it be that the snow exerts that feedback but that some of the jet index variability comes from the atmospheric initial conditions? Some comments are warranted.

Response:

Yes, we tried to address the issue of causality in the previous responses to your comments. It seems that we did not fully clarify this issue last time. We are sorry about that and are making more efforts to address this important issue.

We agree with the reviewer that the jet index variability in observations (black line in Fig. 4a) may arise not only from the lower boundary conditions (i.e., snow cover) but also from the internal atmospheric variability. In other words, both the TPSC-forced atmospheric response and the initial internal atmospheric variability may contribute to the TPSC variability and the downstream jet variability (shadings in Fig. 2 and black line in Fig. 4a). To isolate the TPSC-forced atmospheric response (feedbacks exerted by snow) and to demonstrate that the composites contain causality of relationship and to further reveal the mechanism of interest, we performed a series of numerical experiments. The only difference at the initial time step of the model are the snow cover conditions over the TP for ExpPOS and ExpNEG.

To eliminate the impacts of atmospheric variability embedded in the initial conditions, we carried out several runs for each experiment with different initial dates and conducted an ensemble mean. Since the initial date can be considered as random, the internal atmospheric variability in the initial time step could be largely smoothed out after the ensemble average is applied. Comparisons of ensemble mean results between ExpPOS and ExpNEG could efficiently reveal the TPSC-forced atmospheric responses.

We have incorporated the above discussion into the revised manuscript. Please see Lines 135–142, Lines 378–380 and Lines 384–387 in the revised manuscript.

OTHER MAJOR COMMENTS

Regarding how the land conditions are initialised, is it only the snow cover that is changed between the two experiments (i.e. what about the other land variables such as snow temperature, density, subsurface temperature and so forth)? This should be made clearer in the text.

Response:

In the land surface scheme, the physical snow depth (h_{sno}) and snow water equivalent (m_{sno})

are the direct variables related to the snow cover fraction (f_{sno}). To discuss and clarify the effects of TPSC, we only modified h_{sno} and m_{sno} in the initial boundary conditions for the sensitivity experiments.

Briefly speaking, the target of the experimental design is to generate increased/decreased TPSC in ExpPOS/ExpNEG that are consistent with the IMS composite. The h_{sno} is derived from the f_{sno} in the IMS composite by using the parameterized function from Dickinson (1993). The snow density (ρ_{sno}) is estimated as 100 kg m^{-3} for ExpPOS to emulate the fresh snow increase in positive anomalous TPSC events, and the ρ_{sno} is estimated as 350 kg m^{-3} for ExpNEG to emulate the aged snow remaining in negative anomalous TPSC events. The m_{sno} is then calculated from the modified h_{sno} and the estimated ρ_{sno} . Except for h_{sno} and m_{sno} , other variables at the initial time step remain unchanged.

We have incorporated the above clarification into the revised manuscript. Please see Lines 354–374 in the revised manuscript.

Reference

Dickinson, R. E., Henderson-Sellers, A. & Kennedy, P. J. Biosphere-atmosphere Transfer Scheme (BATS) Version 1e as Coupled to the NCAR Community Climate Model. *NCAR Technical Note* NCAR/TN-387+STR, doi:10.5065/D67W6959 (1993).

In particular, I wonder if the IMS data be a slight overestimation of the snow cover. The authors use the 24km-resolution IMS binary data. I believe that if a 24km² grid box is snow covered (value 1) in IMS, it means that 50% of the area is snow covered, and it is not (value 0) if less than 50% of the grid box is covered. The original data has a pixel resolution of 4km. Although the agreement of IMS with station snow cover (after conversion from snow depth) in Fig S8 is convincing, a brief description of how the original IMS is handled, is warranted. In particular, I wonder if the snow cover extent derived from IMS consistent with other studies, e.g., based on MODIS satellite snow cover? A sentence or two on other recent studies might be worth here.

Response:

1. Thanks for your comments. We agree with you that there might be some systematic bias in the IMS data. Yang et al. (2015) reported an overestimation in the IMS data over the TP. Despite the systematic bias, the IMS snow cover analysis can capture the general subseasonal variability of TPSC, as shown in our evaluation. The IMS snow cover analysis is provided in 24-km and 4-km resolutions. The 24-km resolution dataset and 4-km resolution dataset begin

from February 1997 and January 2004, respectively. We chose the 24-km dataset because of its longer collection period. There are 7658 grid points over the TP; the resolution should be enough for our regional-scale study.

We give a brief description of how the original IMS is handled in Lines 93–114 in the revised supplementary material (Supplementary Note 3):

“The Interactive Multi-Sensor Snow and Ice Mapping System (IMS), which is provided by the National Oceanic and Atmospheric Administration (NOAA), is an interactive system that is used to examine satellite images and other sources of data on snow cover and to generate maps of snow cover distribution^{1–3}. The visible and infrared spectral data from the Polar Operational Environmental Satellites (POES) and Geostationary Orbiting Environmental Satellites were primarily used to generate snow cover data. Moderate Resolution Imaging Spectrometer (MODIS) imagery was used as well. In addition, ground weather observations from many countries were used. Since the visible and infrared data suffer from persistent cloud cover, which makes observations difficult, microwave products from SSM/I (Special Sensor Microwave Imager) and AMSR-E (Advanced Microwave Scanning Radiometer for EOS) are being used in the IMS product. The IMS system also includes the model output from Snow Data Assimilation System (SNODAS) and station-mapped products. The spatial resolutions of visible, infrared, microwave and SNODAS products used in the IMS System vary from 1 to 40 km.”

“The IMS product was manually created by NOAA NESDIS (The National Environmental Satellite, Data, and Information Service) satellite-product group analysts looking at all available satellite imagery, automated snow mapping algorithms, and other ancillary data. The IMS analysts use these multiple sources with different spatial resolutions within the interactive multisensor snow mapping system and re-gridded it to map snow at a 24-km spatial resolution. The analyst begins charting using the map from the previous day, then uses the satellite inputs accordingly. The IMS system allows for faster processing time to produce snow cover maps from satellite remote sensing data.”

2. As the above description indicates, the IMS snow cover product is derived from multiple sources, including MODIS. The daily MODIS snow cover data is absent under cloud cover. One of the important advantages of the IMS snow cover is that microwave products are being used to allow a view through clouds. The daily IMS snow cover product is suitable for larger scale analyses and applications, with the advantage over MODIS of allowing for mitigation of cloud cover. The daily IMS snow cover products without cloud obscuration are prone to be more suitable to investigate the spatial distribution of daily snow cover.

References

1. Helfrich, S. R., McNamara, D., Ramsay, B. H., Baldwin, T. & Kasheta, T. Enhancements to, and forthcoming developments in the Interactive Multisensor Snow and Ice Mapping System (IMS). *Hydrol. Process.* **21**, 1576–1586, doi:10.1002/hyp.6720 (2007).
2. National Ice Center. IMS Daily Northern Hemisphere Snow and Ice Analysis at 1 km, 4 km, and 24 km Resolutions, Version 1. [24 km Resolutions]. *Boulder, Colorado USA. NSIDC: National Snow and Ice Data Center.* doi:10.7265/N52R3PMC (2008, updated daily).
3. Chen, C., Lakhankar, T., Romanov, P., Helfrich, S., Powell, A. & Khanbilvardi, R. Validation of NOAA-Interactive Multisensor Snow and Ice Mapping System (IMS) by Comparison with Ground-Based Measurements over Continental United States. *Remote Sens.* **4**, 1134–1145, doi:10.3390/rs4051134 (2012).
4. Yang, J. *et al.* Evaluation of snow products over the Tibetan Plateau. *Hydrol. Process.* **29**, 3247–3260, doi:10.1002/hyp.10427 (2015).

MINOR COMMENTS

Figure S4 contains little information; it could be more instructive to show the climatology as background contours in Figure 2, to see how anomalies reinforce or weaken climatological winds.

Response:

We have revised Fig. 2 and removed the original Fig. S4. The background contours in the revised Fig. 2 now show the climatological mean of the wintertime 300-hPa zonal wind. It is truly more instructive to see how anomalies modulate climatological winds.

L 109: “In contrast,..”. This sentence is not necessary.

Response:

Thank you. We have omitted that sentence.

L131: remove “experiments.”

Response:

Thank you. We have done this.

L188: the 2nd “downward” should be “upward”.

Response:

Yes. We have corrected it.

L201 : Has “SH” be defined in the text ? (or only in the Table)

Response:

Thanks for your reminder. We have revised the related statements. The SH, which is the abbreviation of sensible heat flux, now only appears in Table 1. In other parts of the manuscript, we use the full “sensible heat flux”.

L214: the cooling response. The reference to Senan et al. should be mentioned here since it is a similar finding.

Response:

We thank the Reviewer and we agree that Senan et al. (2016) is an appropriate reference here. Please see Lines 244–245 in the revised manuscript.

L219-221: the word “adaptative” is not appropriate here. The word “affect” is used twice in the same sentence. Rather use : The zonal wind is in geostrophic balance with geopotential heights (?)

Response:

We thank the Reviewer. We have reworded this sentence following your suggestion. Please see 252–253 in the revised manuscript.

Caption of S9: “simultaneous composites of the daily anomalous TP snow-cover probabilities for the IMS snow cover analysis.” That wording is quite unclear.

Response:

We thank the Reviewer. The sentence has been revised as “The composites of the daily anomalous TP snow-cover probabilities at a lag of 0 days”. We hope it is clear.

Reviewer #2 (Remarks to the Author):

Comment on “Influence of the Tibetan Plateau snow cover on East Asian atmospheric circulation on medium-range time-scales” by Li et al.

General comments

The revised manuscript has sufficiently addressed my concerns on its scientific content. I recommend it to be published after the following minor comments are addressed.

Minor comments

1.The mechanism linked to effects of TPSC variability on East Asian atmospheric circulation at subseasonal scale is still need described clearly.

Response:

Thanks for the comments on how to improve our writing. 1. We tried to clarify why we performed numerical experiments: To isolate the TPSC-forced atmospheric response and to further reveal the mechanism. Please see Lines 135–142 in the revised manuscript. 2. The aim of section “Land surface energy budget and its atmospheric effects” is to investigate the physical process behind the relationship (Lines 191–263). A few corrections have been set in this part. 3. We also paid more attention to clear up the mechanism in the Discussion section. A brief overview of the mechanism has been added. Please see Lines 273–279 in the revised manuscript.

2.The abstract should be further summarized to show the highlight in this research.

Response:

We thank the Reviewer for his/her suggestion. We have highlighted the innovative findings of this study in the abstract. Please see Lines 29–32 in the revised manuscript.

3.The uncertainties in results analysis should be discussed.

Response:

We thank the Reviewer and agree that we should improve the discussion. We have modified the Discussion section. Please see Lines 281–289 in the revised manuscript.

Reviewer #3 (Remarks to the Author):

Authors have addressed most of my questions. However they still need to work on the following.

1, L50-54, add some very related references:

Wang, M., Wang, J., Duan, A., Liu, Y., & Zhou, S. (2018). Coupling of the quasi-biweekly oscillation of the Tibetan Plateau summer monsoon with the Arctic Oscillation. Geophysical Research Letters, 45. <https://doi.org/10.1029/2018GL077136>

Liu Y M, Wang Z Q, Zhuo H F, Wu G X. Two types of summertime heating over Asian large-scale orography and excitation of potential-vorticity forcing II. Sensible heating Over Tibetan-Iranian Plateau. Science China Earth Sciences, 60(4): 733–744, doi: 10.1007/s11430-016-9016-3

Wu G X, Zhuo H F, Wang Z Q, Liu Y M. 2016. Two types of summertime heating over the Asian large-scale orography and excitation of potentialvorticityforcing I. Over Tibetan Plateau. Science China Earth Sciences, 59 (10): 1996–2008, doi: 10.1007/s11430-016-5328-2

Bao Qing, Jing Yang, Yimin Liu, Guoxiong WU and Bin Wang, 2010: Roles of anomalous Tibetan Plateau warming on the severe 2008 winter storm in Central-Southern China. Mon. Wea. Rev., 138: 2375-2384

Liu YM, B. J. Hoskins, M. Blackburn, 2007: Impact of Tibetan Orography and Heating on the Summer Flow over Asia. J. Met. Soc. Japan, 85B: 1-19.

Response:

We thank the Reviewer. Indeed, the suggested studies are very relevant and helpful. They have been included as references in our Introduction. Please see Line 43 and Line 54 in the revised manuscript.

2, Fig. 3

The original Fig. 1 is good for understanding why the snow happened and should be kept or in Supplementary information. The wind signal over the western of the TP should be the key factor to induce the strong snow, and this western wind anomaly are with a barotropic structure (Yu et al., 2011). You can check this in the observation. Please remove Fig. 3d, draw leg -1+0 and set it as in Fig. 3a.

Yu Jingjing, Liu Yimin, Wu Guoxiong, An analysis of the diabatic heating characteristic of atmosphere over the Tibetan Plateau in winter II: interannual variation. Acta Meteor. Sinica. 69: 89-98

Response:

1. Thanks for your comments. The original Fig. 1 showed the 300-hPa zonal wind (U300) responses to the TPSC at a lag of 6 days in both reanalysis composites and numerical experiments. Following Reviewer #1’s advice in the last round of review, we replaced the original Fig. 1 with Fig. 2c and Fig. 3c in the revised manuscript in the last round of revisions to give the reader an overview of the subseasonal evolution. These changes did not affect any results. We would prefer to show Fig. 2c and Fig. 3c only.
2. We thank the reviewer. Yes, we agree that the wind signal over the western TP should be the key factor inducing the TPSC variability. The wind anomalies have a barotropic structure in the observations (Fig. R3-1), as demonstrated by Yu et al. (2011). The wind signal appears in the composites over the western TP because the composites (shadings in Fig. 2 and black line in Fig. 4a) actually combine the two parts of atmospheric variability, the TPSC-forced atmospheric response and the initial internal atmospheric variability, which are responsible for both the TPSC variability and the downstream jet variability. To isolate the TPSC-forced atmospheric response (feedback exerted by snow), as well as to prove that the composites contain a causal relationship, we performed numerical experiments.

As indicated by the reviewer, understanding the factors that generate the TPSC anomalies is an important but challenging topic. In this short article, we mainly focus on the atmospheric responses to the subseasonal signals of TPSC. Actually, what causes the subseasonal TPSC variability is our ongoing work will be shown somewhere in the near future (hopefully).

Figure R3-1 | The composite of the zonal wind in the reanalysis with respect to the TPSCI over the western TP (averaged between 70–85°N). The unit is m/s. Grey areas indicate the terrain elevation.

Reviewers' Comments:

Reviewer #1:

Remarks to the Author:

With this second revision of the manuscript, the authors have addressed all my comments. I recommend publication of the manuscript.